# A Model-Driven-to-Sample-Driven Method for Rural Road Extraction

**Jiguang Dai** [1,2], **Rongchen Ma** [1,2,*], **Litao Gong** [1,2], **Zimo Shen** [1,2] and **Jialin Wu** [1,2]

1   School of Geomatics, Liaoning Technical University, Fuxin 12300, China; daijiguang@lntu.edu.cn (J.D.); 1704070306@stu.lntu.edu.cn (L.G.); 472020770@stu.lntu.edu.cn (Z.S.); 471920571@stu.lntu.edu.cn (J.W.)
2   Institute of Spatiotemporal Transportation Data, Liaoning Technical University, Fuxin 12300, China
*   Correspondence: 471820526@stu.lntu.edu.cn; Tel.: +86-183-4280-0951

**Abstract:** Road extraction in rural areas is one of the most fundamental tasks in the practical application of remote sensing. In recent years, sample-driven methods have achieved state-of-the-art performance in road extraction tasks. However, sample-driven methods are prohibitively expensive and laborious, especially when dealing with rural roads with irregular curvature changes, narrow widths, and diverse materials. The template matching method can overcome these difficulties to some extent and achieve impressive road extraction results. This method also has the advantage of the vectorization of road extraction results, but the automation is limited. Straight line sequences can be substituted for curves, and the use of the color space can increase the recognition of roads and nonroads. A model-driven-to-sample-driven road extraction method for rural areas with a much higher degree of automation than existing template matching methods is proposed in this study. Without prior samples, on the basis of the geometric characteristics of narrow and long roads and using the advantages of straight lines instead of curved lines, the road center point extraction model is established through length constraints and gray mean contrast constraints of line sequences, and the extraction of some rural roads is completed through topological connection analysis. In addition, we take the extracted road center point and manual input data as local samples, use the improved line segment histogram to determine the local road direction, and use the panchromatic and hue, saturation, value (HSV) space interactive matching model as the matching measure to complete the road tracking extraction. Experimental results show that, for different types of data and scenarios on the premise, the accuracy and recall rate of the evaluation indicators reach more than 98%, and, compared with other methods, the automation of this algorithm has increased by more than 40%.

**Keywords:** rural road extraction; model-driven; sample-driven; line sequence; template matching methods; HSV color space; improved line segment histogram; panchromatic and HSV space interactive matching model

## 1. Introduction

Road extraction from remote sensing images plays an incredibly important role in map updating, emergency responses, smart cities, sustainable urban expansion, vehicle management, urban planning, traffic navigation, public health, drone navigation, disaster management, agricultural development, driverless vehicle routing, and traffic management [1–6]. With the development of remote sensing technology, remote sensing images have become easier to obtain [7]. Therefore, using remote sensing images to complete road extraction has become popular. For the practical road extraction algorithm using remote sensing image to complete road engineering, the results are as follows: (1) high precision and recall, (2) high degree of automation, and (3) the results can be vectorized.

With the development of modernization, the informatization of rural roads is an inevitable requirement of the development of the times. Compared with urban roads, rural roads occupy a higher proportion of the total road mileage. For example, in China, the total mileage of rural roads has reached 4.2 million kilometers, accounting for 83.8%

of the country's total road mileage. In order to accelerate the economic development of rural areas, the storage of vectorized data on rural roads has become an important task. However, due to the capital, planning, environmental, and other conditions involved the construction of rural roads, compared with those of urban roads, rural roads exhibit the characteristics of irregular curvatures, narrow widths, and diverse materials. For these roads, very few methods exist for targeted research. The common traditional algorithms used in rural roads have poor robustness and low degree of automation, and the results of deep learning algorithm are usually difficult to vectorize and cannot directly meet the production requirements of data storage. Therefore, a method suitable for the practical application of rural road engineering has not been found yet.

In this paper, combining the characteristics of rural roads, under the premise of ensuring accuracy, with the purpose of improving the degree of automation of rural road extraction, combined with actual production needs, the study of rural road extraction methods is carried out.

The rest of this paper is structured as follows: The related work is described in Section 2. Section 3 illustrates experimental data and the proposed method in detail. Section 4 presents the experimental setup and the comparison of road extraction results. Discuss the advantages of the proposed method in Section 5. Finally, Section 6 provides the conclusion and the future work.

## 2. Related Work

Recent years, many scholars have carried out research on road extraction methods based on remote sensing images from different perspectives. According to the requirement of prior samples, we divide road extraction methods into model-driven methods and sample-driven methods.

Model-driven methods do not require prior samples but directly use the differences between different objects in feature space to construct theoretical models for road extraction. This paper classifies model-driven methods according to geometric features and texture features.

Many scholars consider the homogeneity of the spectral texture inside the road and use the texture feature to complete the road extraction work, where a classic method is the object-oriented method [8]. Based on the homogeneity in the spectral texture and edge features of a road, an image segmentation algorithm is used to cluster pixels with similar features to form segmentation region units. On this basis, the road is extracted through geometric spectral analysis. The selection of the segmentation method, such as the common models of threshold segmentation [9], multiscale segmentation [10,11], fuzzy C-means [12], graph segmentation [13], and edge segmentation [14], is the core problem. For example, Maboudi et al. [15] used a multiscale model, combining color and shape information to segment images; classified segmentation units by using structure, spectrum, and texture features; and connected road fault zones by the tensor voting method. However, the method of segmentation and acquisition of image objects does not consider the high-level features of the image, such as morphological information and contextual semantic information, and includes pixel aggregation based on spectral features, without making full use of other features of high-resolution remote sensing images [16,17]. Therefore, the object unit obtained by the segmentation method often does not match the shape of the actual target feature, and, as a result, the processing results of the object-oriented method cannot be converted into results with actual geographical significance [18,19].

Based on the geometric features of roads, some scholars choose to use the narrow and long geometric features of roads to extract them. Among these features, parallel edges [20,21], line segments [22], and path morphologies [23] are used to extract roads by using geometric features. However, this method of road extraction based on geometric features places a high limit on the edges of roads in images, and roads with curves, weak edges, and occlusions often do not meet the requirements of this method. In general, although the model-driven method has a high degree of automation, its extraction accuracy and com-

pleteness are still unsatisfactory in the face of a complex road environment, and due to the lack of human intervention, the vectorization of the results requires extensive intervention.

In sample-driven methods, the a priori samples of ground objects are selected manually, and the model is established to determine the fitting of the feature parameters of ground objects. On the basis of these parameters, the discriminant function is used to analyze the pixels to identify the ground objects. Deep learning is a typical sample-driven method. This method learns from training samples and sets an end-to-end parameter model. The initial parameter model is constantly optimized, and the test samples are identified through the discriminant function and judgment function. In recent years, deep learning methods have been widely used in road extraction. Mnih and Hinton [24] proposed a road extraction method based on a restricted Boltzmann machine (RBM) for aerial images for the first time by using deep learning technology. In 2015, Long et al. [25] proposed the fully convolutional network (FCN), which is the most commonly used road extraction architecture but needs well-annotated samples to train these deep learning models. Since then, various FCN-like architectures have been proposed, including U-Net [26], SegNet [27], and DeepLabV3+ [28]. In this direction, Panboonyuen et al. [29] improved the road extraction accuracy by using SegNet network, combined with exponential linear unit (ELU) function; landscape metrics (LMs) to further reduce the misclassification; and, finally, conditional random fields (CRFs) to sharpen the extracted roads. The proposed method improved the integrity of road extraction results. However, most of these methods are encoder–decoder structure: in the part of decoder, the boundary accuracy will be reduced, resulting in the discontinuity of road extraction results. In order to solve the problem of discontinuous extraction results, Gao et al. [30] proposed a postprocessing method for roads with broken connections. However, given the changeable nature of image condition, the postprocessing operations are complex, which reduces the automaticity of road extraction. Zhou et al. [31] proposed a road extraction network based on boundary and topology perception. The deep learning method can usually achieve more than 90% accuracy in road extraction results. However, the integrity of deep learning road extraction results is usually less than 85%, and the road extraction results are only two or more classification problems. There are no vector topological relationships between the classification results, and the data cannot be directly put into a database. Much manual postprocessing is needed; otherwise, the actual application requirements cannot be met [32,33]. In addition, when there are differences between the training sample sets and test sets, it is difficult for deep learning to realize transfer learning, which means different sample sets need to be selected for road areas with very different characteristics [34,35].

The template matching method is also a simple, sample-driven method. In this method, local effective samples are formed through manual input of initial information, the parameter information of local samples is learned as the benchmark of matching and tracking, and roads are extracted by human–machine interaction. Classic templates include section templates [36], rectangular templates [37], T-shaped templates [38], circular templates [39], and sector descriptors [40,41]. For example, Zhang et al. [42] manually input three points on the edges of both sides of a road, constructed a rectangular template and input direction parameter information, and completed the extraction of the road through an iterative method. Different from the deep learning method, which requires a large number of training sample sets, this type of method is based on part of the sample training and has good feature similarity with adjacent features and strong human intervention. Therefore, it has the advantages of high extraction accuracy (both the precision and recall can reach more than 95%) and strong practicability (the extracted road information can be vectorized for easy storage). However, with shortcomings in the remote sensing images of complex scenes, the template matching method requires extensive manual intervention.

Therefore, on the basis of the template matching method, to improve the degree of automation and precision of rural road extraction, we sought to combine the advantages of the high degree of automation of a model-driven method and the high precision of a sample-driven method and propose a model-driven-to-sample-driven road extraction

method for rural road extraction. In summary, the main contributions of the algorithm proposed in this paper are as follows:

(1) The results can be vectorized into a library. Deep learning models require a large number of prior sample sets [43,44], which is prohibitively expensive and laborious. In addition, the road extraction results of deep learning methods lack topological network information, and the results still need extensive intervention before the data can be stored in a database [31,45]. In this study, we first developed a model-driven method to automatically extract some rural roads. Second, we adopted a method of human–computer interaction to complete road extraction. The extracted road results included topological information, which meets the requirements of practical engineering.

(2) High degree of automation. The existing extraction methods face a contradiction between automation and quality [46,47]: the higher the degree of automation, the more difficult it is to control the quality of road extraction; in contrast, the lower the degree of automation, the more difficult it is to guarantee the accuracy. We fully exploited the high precision of the template matching method and improved the template matching method to reduce manual participation. Additionally, on the basis of ensuring accuracy and integrity, a model-driven method with a high strength constraint was introduced to make the algorithm more automatic. This makes our method not only have the characteristics of high precision and high recall but also improves the degree of automation by more than 40% through the calculation of input seed points, which solves the problems of low degree of automation of template matching and the requirement a lot of human participation.

(3) High compatibility. We fully consider that rural roads exhibit irregular curvature changes, narrow widths, and diverse pavement materials in the algorithm design process by improving the multiscale line segment orientation histogram (MLSOH) model and using a full-color and the hue, saturation, value (HSV) spatial interactive matching model to improve the adaptability of traditional template matching methods. We demonstrated this feature in the actual scene experiment. In the three experimental images, a total of more than 50 kilometers mountain roads and forest roads were used to represent irregular curvature transformations and narrow widths. At the same time, a total of more than 30 kilometers of roads at junctions between small towns and forest areas, roads in rural villages, and roads between farmland, shows the characteristics of pavement material diversification. The extraction precision and recall of all types of roads could reach more than 98%. We proved the algorithm proposed in this paper has good extraction effect in mountain road, forest road, small town road, and rural village road.

## 3. Materials and Methods

### 3.1. Experimental Data

Our research data are high-resolution orthophoto panchromatic remote sensing images, with resolution of less than 1 meter, and the corresponding multispectral images. First, we selected GeoEye-1 images from Australia from February 2009 as the experimental data. Second, we selected remote sensing images of GF-2 in Hubei Province, China, from August 2019 and Pleiades satellite remote sensing data from Liaoning Province, China, from January 2016. The specific conditions are given shown in Table 1.

### 3.2. Methodology

The technical route of our method for efficiently and accurately extracting rural roads is shown in Figure 1a. In Section 3.2.1, the preprocessing method is introduced. First, the line sequence is extracted on the basis of the panchromatic remote sensing image to obtain the structural characteristics of the road. Then, L0 filtering [48] processing is performed on the panchromatic and multispectral images to improve the homogeneity of the road interior. Finally, the panchromatic and multispectral images are fused, and the fused multispectral images are converted to HSV color space. (b) In Section 3.2.2, the model-driven approach is introduced. Through the comparative analysis of the length of the line sequence and the difference in the gray mean, the road center point extraction model is

established, and, finally, topological connection analysis is performed on the obtained road center point. (c) In Section 3.2.3, the sample-driven method is used to improve the MLSOH model based on the line sequence to complete the prediction of the local road direction, to establish a multicircle matching template, and to complete tracking matching through the panchromatic and HSV spatial interactive matching model. Figure 1 shows a flowchart of the proposed method.

**Table 1.** Technical specification of different remote sensing image used in this study.

| Data Type | Spectral Region | Band Range (nm) | Spatial Resolution (m) |
|---|---|---|---|
| Geoeye-1 | Panchromatic | 450–800 | 0.41 |
| | Blue | 450–510 | 1.65 |
| | Green | 510–580 | 1.65 |
| | Red | 655–690 | 1.65 |
| GF-2 | Panchromatic | 450–900 | 1 |
| | Blue | 450–520 | 4 |
| | Green | 520–590 | 4 |
| | Red | 630–690 | 4 |
| Pleiades | Panchromatic | 470–830 | 0.5 |
| | Blue | 430–550 | 2 |
| | Green | 500–620 | 2 |
| | Red | 590–710 | 2 |

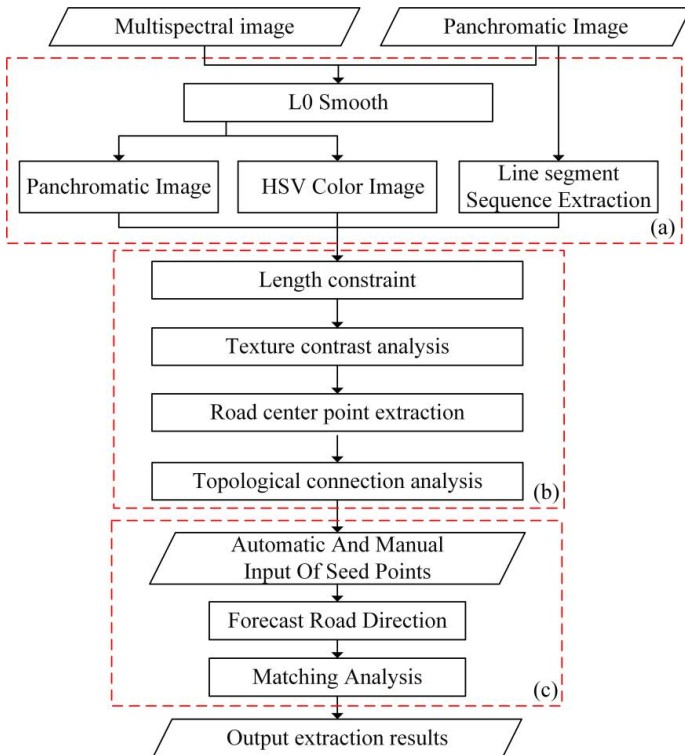

**Figure 1.** Method flowchart.

### 3.2.1. Preprocessing

As artificial objects, roads present obvious structural and textural homogeneity in images. Therefore, the purpose of our preprocessing is to extract the road structure line sequence, which can highlight the expression of the road, and then use the L0 filtering method to smooth panchromatic and multispectral images to enhance the homogeneity of the road and the difference between the inside and outside of the road. Finally, the panchromatic and multispectral images are fused, and the fused multispectral images are transformed into HSV color space images.

(1) Line sequence extraction

Due to the continuity of Canny edge tracking results, the edge points with 8 connected relations were grouped using the edge points extracted by Canny, and the same grouped edge was one connected component. As shown in Figure 2b, the same edge group is given the same color for display. On this basis, line segments are extracted by using the Dai's method [49]. Since the results of line segments are independent, in order to clearly express the differences between line segments, as shown in Figure 2c, we used color to distinguish line segments and assigned different colors to different line segments for display. Finally, the line segments in the same group were processed to form different line segment sequences.

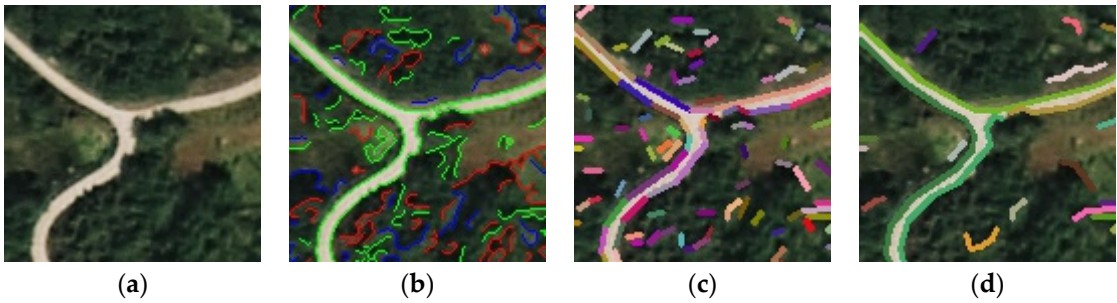

(**a**) (**b**) (**c**) (**d**)

**Figure 2.** Line sequence extraction different steps result image. (**a**) Original image; (**b**) edge image; (**c**) line segment extraction result; (**d**) line sequence extraction result.

On the basis of Figure 2b,c, we took line segments $l_i$ and $l_j$ to form line sequence *ful_l$_{ij}$* as an example. In this process, the line segments $l_i$ and $l_j$ need to satisfy the following conditions:

(1) Phase constraint. $l_i$ and $l_j$ belong to the same object edge, and the phase difference between line segments should be small.

$$|phase(l_i) - phase(l_j)| \leq \tau_{phase} \qquad (1)$$

where $\tau_{phase}$ represents the phase constraint threshold of two line segments and can usually be set to 2 radians.

(2) Distance constraint. The image plane distance between $l_i$ and $l_j$ should be small. In this paper, the minimum Euclidean distance between the end points of lines $l_i$ and $l_j$ is constrained as follows:

$$Min\_Dis(EDP(l_i) - EPD(l_j)) \leq \tau_{EPD} \qquad (2)$$

where *Min_Dis* is the shortest Euclidean distance, $EDP(l_i)$ represents the coordinates of any end point of line $l_i$, $EPD(l_j)$ represents the coordinates of any end point of line $l_j$, $\tau_{EPD}$ is the distance reference threshold, and $\tau_{EPD}$ is set to 2 pixels.

When the line segments $l_i$ and $l_j$ satisfy Equations (1) and (2), the end points of one line segment are updated to achieve adjacency. As shown in Figure 2d, we extract the line sequence of the edge of the mountain road. We can see that the line sequence better expressed the edge of the curved road.

(2) L0 filtering

In high-resolution images, the surface brightness of road images is uneven. For example, the complexity of road surface extraction is increased by the factors of the surrounding objects and shadow occlusions. From the perspective of the image gradient, the L0 smoothing filter [48] can smooth most of the low noise while maintaining the important edge information to the maximum extent. Therefore, we used the L0 smoothing filter to remove nonzero gradients, to preserve road boundary information, to eliminate unimportant details, and to enhance the saliency of the image. $C(S)$ (measurement of the

L0 norm of the image gradient) is a gradient measurement that does not depend on the gradient itself. Therefore, if only the contrast is changed, the result is not affected.

$$C(S) = \#\{p||\partial_x S_p| + |\partial_y S_p|\} \tag{3}$$

where the gradient of $S_p$ is used to calculate the color difference between each pixel $p$ and the adjacent pixels in the x- and y-directions. #{} denotes a statistical operation used to count the number of signals p satisfying that $|\partial_x S_p| + |\partial_y S_p|$ is not equal to zero, that is, the L0 norm of the image gradient.

As shown in Figure 3, the L0 filter is used to remove the noise inside the road, improve the homogeneity inside the road, and increase the difference inside and outside the road.

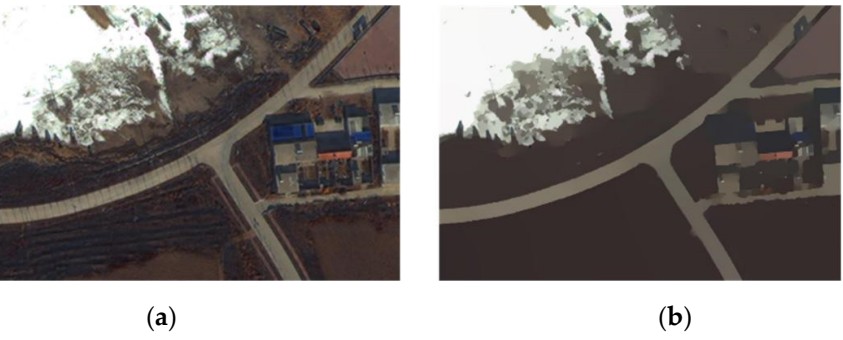

(**a**)  (**b**)

**Figure 3.** Comparison of the L0 filtering effect. (**a**) Original image; (**b**) after filtering.

(3) HSV color space conversion

Compared with panchromatic images, multispectral images have more abundant color information and can display all the color features of ground objects. We used a pansharpening model in ENVI (The Environment for Visualizing Images) to fuse low-spatial-resolution multispectral images with high-spatial-resolution single band panchromatic images to generate high-resolution multispectral remote sensing images. However, in RGB color space, the R, G, and B components are linearly correlated, and it is difficult to correctly represent the actual perceived color differences with the Euclidean distances between different colors. The HSV color space, also known as the hexcone model, was created by A. R. Smith in 1978. According to the intuitive characteristics of color, it can clearly separate chroma, saturation, and brightness and reduce the influence of light. Compared with the RGB color model based on the Cartesian coordinate system, the HSV color space is relatively independent, which is more in line with human perception characteristics [50].

As shown in Figure 4, the sand and gravel pavement of the rural road, in the multispectral image, the color difference between the blue building and the road is significant through human visual perception, but in the panchromatic image, the two colors are similar and cannot be effectively distinguished. Therefore, on the basis of panchromatic image, we used HSV color space to improve the accuracy of road matching.

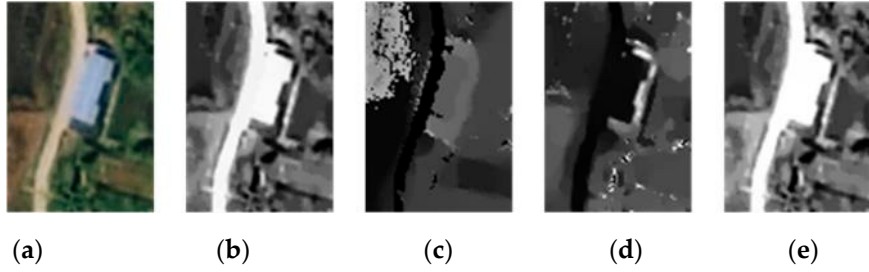

(**a**)  (**b**)  (**c**)  (**d**)  (**e**)

**Figure 4.** Image. (**a**) Original RGB image; (**b**) gray image; (**c**) hue image; (**d**) saturation image; (**e**) value image.

### 3.2.2. Model Driven Approach

A model-driven method provides an abstract representation of the features of a ground object [51,52]. Therefore, we proposed a model-driven method that included geometric, gray mean, and topological features [53]. First, we analyzed the extraction results of line sequences for the significant linear edge features of roads in the image. The longer the line sequences are, the greater their probability of becoming road edges. Second, based on the gray mean difference between the road and nonroad regions, we compared and analyzed the gray mean information of the two sides on the basis of longer line sequences. Then, based on the characteristics of the parallel edges on both sides of the road and the homogeneity of the road internal texture, we established the road center point extraction model. Finally, according to the topological characteristics of the road, the topological connection of the extracted road center points was analyzed to improve the integrity of road extraction.

(1) Length constraint

Roads are usually long and continuous, especially in rural areas where there are few other features with long, linear information. Therefore, in the line sequence extraction results, a length greater than *T_len* is regarded as a suspected road edge line sequence, and then the subsequent gray mean and parallel judgment is carried out. As shown in Figure 5, red line sequences are those with lengths greater than *T_len*, and most of them are road edge line sequences.

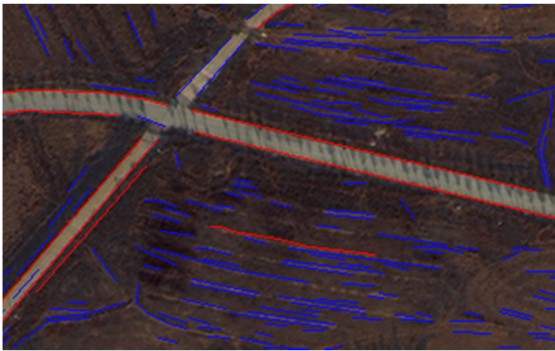

**Figure 5.** Schematic diagram of length constraint results.

(2) Gray mean contrast constraint

In most cases, the same road is made of the same or similar materials and has gray mean information different from that of the surrounding objects. Therefore, we used the gray mean information to complete the gray mean constraint analysis of the road edge line sequence. In Figure 6, the black continuous line segment is a line sequence that meets the length constraint. A rectangular buffer with a width of 3 pixels was established on both sides of each line segment contained in the current line sequence. The buffer was set to 3 pixels, considering that the road width was not less than 2 m. Additionally, we judged whether the left and right sides met the requirements of Equation (4). If the requirements of Equation (4) were met, the line sequence continued to be a suspected road edge line sequence; otherwise, the line sequence did not represent the road edge position.

$$Abs(G_R - G_L) >= G_D \tag{4}$$

where $G_R$ is the mean gray value of all pixels in the red buffer area, $G_L$ is the mean gray value of all pixels in the green buffer area, and $G_D$ is the constraint threshold.

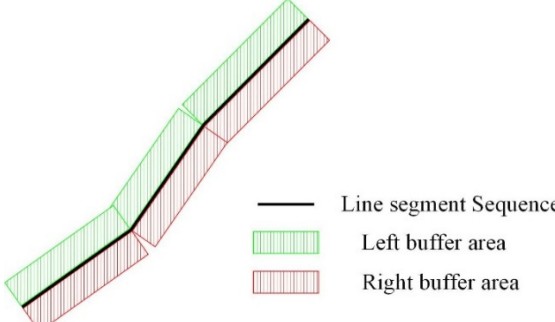

**Figure 6.** Gray mean comparison diagram.

(3) Road center point extraction model

Rural roads are relatively simple and usually have only two relatively parallel road edges. Under the guidance of this idea, on the basis of meeting the first two steps, we first analyzed whether there was a parallel line sequence in the suspected road edge line sequence, as shown in Figure 7. We determined that the blue line sequence to be analyzed was parallel to the red line sequence, and the projection distance $D_S$ between them met the requirements of Equation (5). Second, based on the high homogeneity of the internal road texture, we set the parallel region in Figure 7 as the texture analysis region. When Equation (6) was satisfied, the line sequence was the road edge. When the line sequence satisfied Equations (5) and (6), as shown in Figure 7, the road center point between the parallel line sequences could be extracted, and the projection distance $D_S$ between the line sequences could be set as the road width.

$$D_S \in D_f \tag{5}$$

$$Var <= G_V \tag{6}$$

where $D_S$ is the projection distance between line sequences, $D_f$ is the range of constraint threshold. According to rural road design standards, the width of rural road is usually between 3–15 m; we set $D_f$ as [3/$res$,15/$res$], where $res$ is the spatial resolution of the image; $Var$ is the variance of projection analysis area, and $G_V$ is the overall constraint threshold.

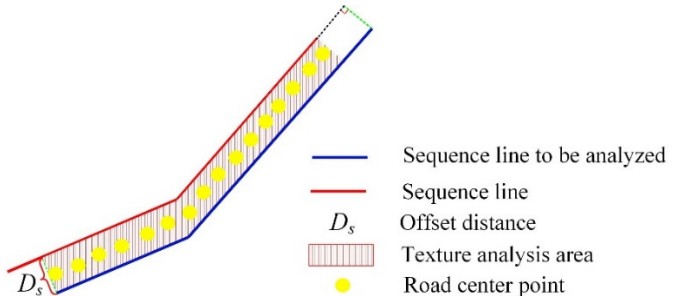

**Figure 7.** Schematic diagram of road center point extraction.

(4) Topological connection analysis

Based on the characteristics of road topology, we analyzes the topological connection, connected and merged multiple road center point sets, and established their topological relationship. As shown in Figure 8, the distance $D_{pp}$ from the end point to the end point of each point set was analyzed. If the distance relationship met Equation (7), the projection distances $D\_sp1$ and $D\_sp2$ from the two end points to the corresponding line segment were determined. If the projection distance satisfied Equation (8), the two point sets were regarded as the center points on the same road; these points can be connected and merged to establish the topological relationship.

$$D_{pp} <= Dis\_p \tag{7}$$

$$D\_sp_1 + D\_sp_2 <= D_f \tag{8}$$

where $Dis\_p$ is the constraint threshold of the distance between points and $D_f$ is the constraint threshold range, the same as the constraint range of Equation (5).

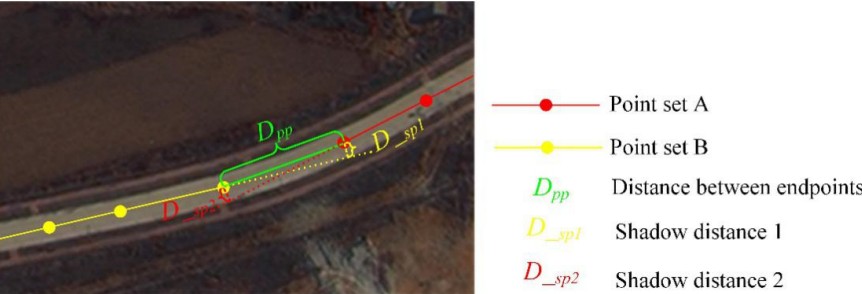

**Figure 8.** Schematic diagram of merging and connecting road center points.

### 3.2.3. Sample Driven Method

We took the road center point obtained by the model-driven method as the initial seed point and set $D_S$ as the road width; when manual point filling was needed in the tracking process, we used the adaptive template method to obtain the road center point and road width [39]. In the tracking process, local direction prediction, multicircle template construction, and panchromatic and HSV space interactive matching were started with a step size of twice the road width, and the extraction of rural roads was completed by iterative tracking. In addition, if an occlusion did not meet the requirements in the tracking process, the step size could be increased to 5 times the road width [40].

(1) Prediction of local road direction

Considering the irregular curvature changes in rural roads, changes in the local road direction can be greater than 120 degrees; if no local road direction prediction is included, the matching point is readily placed outside of the road, which inevitably increases the degree of manual participation. The MLSOH descriptor [40] establishes a road local direction prediction model based on the semantic relationship between the local road direction and the edge direction of neighboring objects and the direction and length of the local range detection line segment. However, in our method, considering the irregular curvature changes in rural roads, it is difficult to ensure the accuracy of direction prediction with the length of a single line segment. Therefore, we modified MLSOH based on the line sequence extraction results.

As shown in Figure 9a, in the process of local road tracking, the green current tracking point is the center, and the side length to establish a red rectangular box is two times the road width. Figure 9b,c show the results obtained by using the line segment extraction method [54] and our line sequence extraction method. The prediction results obtained by the MLSOH model, as shown in Figure 9d, are very different from the actual direction of the current road, due to the interference of other line segments. In this case, we improved the MLSOH model, changing the length of the line segments in the ordinate direction to the length statistics of the line sequence, and the direction is the direction of the line segment in the rectangular box of the line sequence. As shown in Figure 9e, after the MLSOH model was improved, the predicted direction is the actual direction of the road. In Figure 9, the yellow dots represent the existing tracking points, the green dots represent the current matching tracking points, and the red rectangular box represents the local analysis range. Different line segments in Figure 9b are represented by different colors, and different line sequences in Figure 9c are represented by different colors.

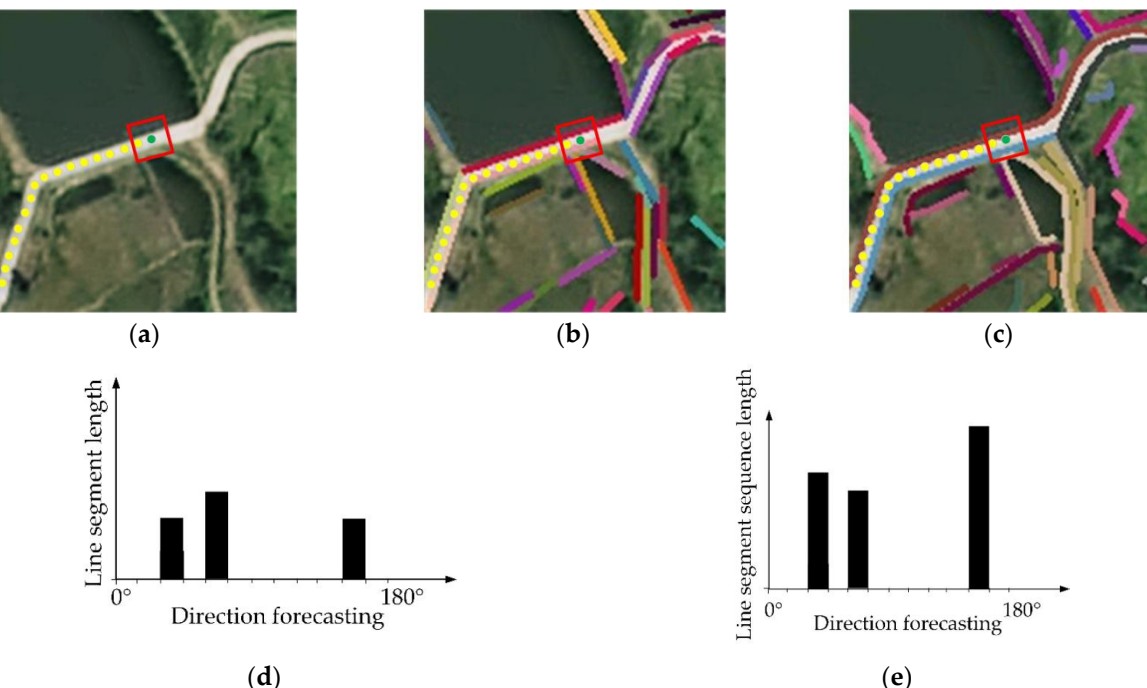

**Figure 9.** Multiscale line segment orientation histogram (MLSOH) model improvement demonstration. (**a**) Road seed point tracking image; (**b**) segment extraction results display; (**c**) line sequence results display; (**d**) MLSOH model prediction results; (**e**) road direction prediction results of this algorithm.

(2) Multicircle Template

Considering that roads with a width greater than 2 m in rural areas can be regarded as rural roads [55] and that they can change direction suddenly from time to time, this paper adopted a circular template as the matching template. To highlight the difference between road and nonroad areas, this paper designed a multicircle template based on of the direction prediction model to adapt to the narrow road width situation. The multicircle matching template takes a reference point $O$ as the center of the circle, sets the step length $S$ as the road width, and obtains seven points ($P_{1-7}$) along the road prediction directions $\theta$, $\theta \pm 10°$, $\theta \pm 20°$, and $\theta \pm 30°$. Then, 7 circular templates are created with a radius of $S/2$.

In Figure 10, the red dot is the reference point, the red circle is the reference template, the green dot is the point to be matched, and the green circles are the circular templates to be matched.

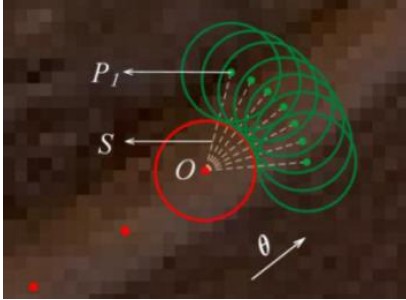

**Figure 10.** Establishment of matching templates.

(3) Panchromatic and HSV space interactive matching model

1. Panchromatic image matching model

Rural road materials include various types of sand and gravel; thus, the spectral characteristic of adjacent ground objects in the panchromatic image are similar to those of roads. To reduce the matching errors caused by the above situation, this paper designed a

template matching measurement model, a template comparison model, and a local analysis model, in that order.

(a) Template matching measurement model. As shown in Equation (9), we calculated the absolute value of the difference between the average gray value of each circular template and the reference template and selected the minimum value as the best matching circular template:

$$Gray_i = \left| \frac{1}{circ_i} \sum_{(m,n) \in cir\_i} G(m,n) - \frac{1}{circ_R} \sum_{(x,y) \in cir\_R} G(x,y) \right| \tag{9}$$

where $Gray_i$ is the matching difference between the *i*-th circular template to be matched and the reference template, $circ_i$ is the number of pixels in the template to be matched, $(m,n) \in cir\_i$ represents all the pixels in the circular template, and $G(m,n)$ is the gray value of the pixel at position $(m,n)$. The variable $circ_R$ represents the number of pixels in the reference template, $(x,y) \in cir\_R$ is the coordinates of all pixels in the reference template, and $G(x,y)$ is the gray value at position $(x,y)$.

(b) Template comparison model. As shown in Equation (10), calculating multiple circular template matching measured both the difference between the maximum $Gray_{i\_max}$ and minimum $Gray_{i\_min}$ to judge whether there was a difference between the road and nonroad positions. If the requirement of Equation (10) was satisfied—that is, the difference between the two was greater than $g$ (the grayscale is equally divided into 16 levels, and the size of each level is $g$)—then we continued to judge from (c) (below); otherwise, we used the HSV color space matching model and tried judging again:

$$|Gray_{i\_max} - Gray_{i\_min}| > g \tag{10}$$

(c) Local analysis model. The consistency of the material along the same road was considered to prevent the matching result from being affected by a local template aberration. We calculated the difference between the gray information of $Gray_{i\_min}$ point and the gray average of the reference template set to satisfy Equation (11) to indicate that the circular template was a matching template:

$$\left| G(a,b) - \overline{G(A)} \right| \leq T_g \tag{11}$$

where $G(a,b)$ is the gray of the $Gray_{i\_min}$ point and $\overline{G(A)}$ is the gray mean value of the reference template set $A$ (the 5 most recently obtained reference templates; when the number of reference templates was less than 5, all the templates were obtained), and $T_g$ is the local constraint threshold.

2. HSV space matching model

This paper started with the template parameter analysis, developed a dynamic weight allocation model, and established an HSV space template matching model.

(a) Template parameter analysis. Each template needed to obtain a gray distribution histogram for each of the hue, saturation, and value modes. However, the values in the three spatial ranges were inconsistent: for example, the interval range of hue was [0,180). Therefore, for convenient calculation, and through experimental verification, the range was linearly normalized to [0,16). As shown in Figure 11, the red circle is the template, the three histograms depict the distributions of the template in the three spatial parameters, the horizontal axis represents the gray value, and the vertical axis represents the number of pixels.

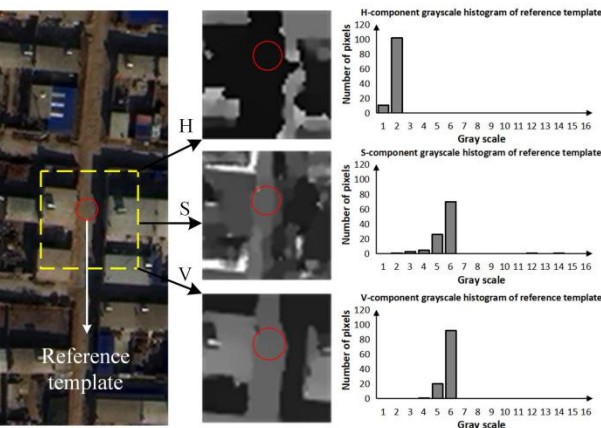

**Figure 11.** Each component the reference grayscale histogram template.

(b) Dynamic weight allocation model. Usually, the weight of each component is not considered in the HSV matching model, but it was difficult to highlight the difference between road and nonroad. Therefore, considering the difference of various ground features in different components of HSV space, on the contrary, the difference of road and road in each component of HSV space was basically the same [56]. The dynamic weight allocation model was designed to enhance the difference between road and nonroad in the HSV matching model. Based on the template parameter analysis result, taking the hue feature as an example, the weight parameter of the hue feature in the matching model is calculated using Equations (12)–(14):

$$H_n = \sum_{i=1}^{i=7} \left| \sum_{j=1}^{j=N} \delta\left[b\left(x_j\right) - n\right] - h_n \right| \tag{12}$$

$$H^* = \sum_{n=1}^{m} H_n \tag{13}$$

$$H^* = \frac{H^*}{(H^* + S^* + V^*)} \tag{14}$$

where $H_n$ is the difference between the 7 templates to be matched and the reference template in the $n$-th interval of the hue feature, $\{x_j\}^N{}_{j=1}$ represents the position of each pixel in the template, $b(\cdot)$ is used to calculate the histogram interval corresponding to the feature value at $x_j$, $\delta$ is the Kronecker function, $\delta[b(x_j) - n]$ is used to determine whether the feature value interval at the pixel $x_j$ in the template to be matched is $n$ (yes 1, not 0), $h_n$ is the number of pixels in the interval $n$ of the reference template, $\left| \sum_{j=1}^{j=N} \delta\left[b\left(x_j\right) - n\right] - h_n \right|$ means the difference in the number of pixels between the template to be matched and the standard template in the interval $n$, and $m$ is the number of intervals. $H^*$ is the matching weight of the hue feature.

(c) HSV space template matching model. We calculated the matching measure $D_i$ between each circular template and the reference template in the multicircular template, and the circular template with the smallest $D_i$ value ($D_{min}$) is the best match. At the same time, if the difference between the minimum value $D_{min}$ and the maximum value $D_{max}$ of $D_i$ satisfies Equation (17), then, it will be recorded as the reference template:

$$DH_i = \left| \frac{1}{circ_i} \sum_{(m,n)\in cir\_i} H(m,n) - \frac{1}{circ_R} \sum_{(x,y)\in cir\_R} H(x,y) \right| \tag{15}$$

$$D_i \sqrt{(H^*) \times (DH_i)^2 + (S^*) \times (DS_i)^2 + (V^*) \times (DV_i)^2} \tag{16}$$

$$|D_{max} - D_{min}| > g \qquad (17)$$

where $DH_i$ is the match value between the *i*-th circular template and the reference template, and $H(m,n)$ is the value at the $(m,n)$ position in the hue layer.

### 4. Experimental Analysis and Evaluation

*4.1. Comparison Method*

To verify the effectiveness and automation of the algorithm, we did not select deep learning road extraction results to ensure the accuracy precision, recall, IoU (Intersection over Union), and F1(F-score) of the road extraction results. Instead, three template matching methods were selected for comparative experiments. First, the completeness of the road extraction results of the deep learning model cannot reach more than 95%, and the extraction results cannot be vectorized. Therefore, we chose the template matching method that could form vector results. These three methods are the T-shaped template of Lin Xiangguo et al. [38], the circular template of Lian Renbao et al. [39], and the sector descriptor method proposed by Dai Jiguang et al. [40]. The T-shape model uses the angle texture feature to accurately locate the initial road point and calculate the width of the road and the direction of the road and uses gray least square matching to extract the center point of the road. The circular template uses the iterative interpolation method to combine the spectral and gray level information to search other road points between the starting point and the ending point to complete the road extraction. Based on the texture difference between the road area and the nonroad mixed area, the sector descriptor method combines triangles to form the sector descriptor and highlights the road image texture features to track the road. The above three methods of comparison are coded in C++ language.

*4.2. Parameter Analysis*

To increase the universality of the parameters, 200 roads were randomly selected to analyze the parameter settings:

(1) Length constraint threshold $T\_len$. According to the geometric characteristics of narrow and long roads, the length constraint parameter was proposed to judge whether a line sequence is a road edge line sequence. As shown in Figure 12, the horizontal axis is the length of the line sequence, and the vertical axis is the probability of whether it is a road edge line. When the length of line sequence $l$ is greater than 100, the probability of a line sequence belonging to a road edge is greater than 85%, and the longer the line is, the greater the probability is. Therefore, the length constraint threshold $T\_len$ was determined to be 100.

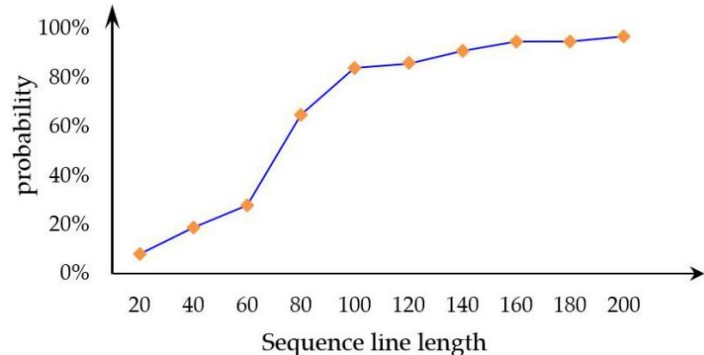

**Figure 12.** Length threshold analysis.

(2) Gray mean analysis constraint threshold $G_D$. According to the gray features on both sides of the road edge, this parameter was proposed to judge the difference between the gray mean values in the buffer areas on both sides of the line sequence. As shown in Figure 13, the horizontal axis is the difference between the road gray mean values on both sides of the line sequence, and the vertical axis is the probability of whether the line is a

road edge line. When the gray mean difference value $G_D$ is greater than 50, the probability of the line being the road edge is greater than 85%, and the greater the gray mean difference is, the greater the probability. Therefore, we determined that the texture analysis constraint threshold $G_D$ should be 50.

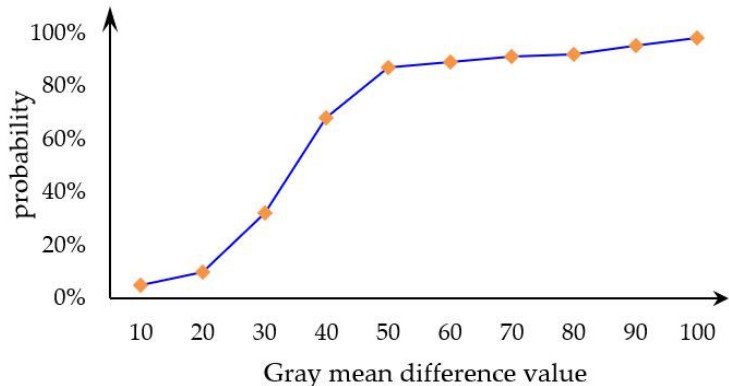

**Figure 13.** Gray mean difference threshold analysis.

(3) Global constraint threshold $G_V$. Considering the material consistency of the same road, the global constraint parameter is determined by calculating the gray standard deviation between the road center points extracted from the parallel line sequences of the road. As shown in Figure 14, the horizontal axis is the gray standard deviation of the same road in different regions of the remote sensing image, and the vertical axis is the number of roads with each gray standard deviation among the 200 roads. The gray standard deviation of the same road was mostly between 3–35, so, considering the maximum compatibility, the global constraint threshold $G_V$ was set to 35.

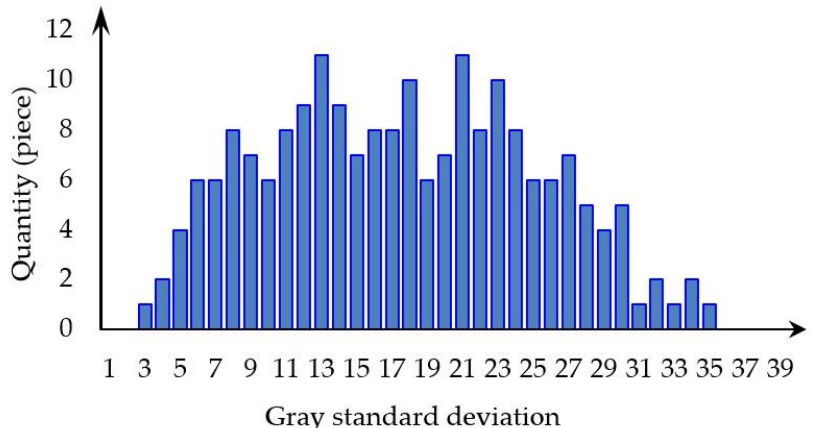

**Figure 14.** Global constraint threshold analysis.

(4) Distance constraint threshold *Dis_p*. This distance constraint parameter considers the topological relationship of the road and is determined by calculating the distance relationship between the road endpoint and the endpoint or node. As shown in Figure 15, the horizontal axis is the distance between the end points, and the vertical axis is the probability that the road is the same when the projection constraint is satisfied. When the distance is less than 60 pixels, the probability that the road is the same is more than 95%. To ensure the accuracy of road extraction, we set *Dis_p* to 60 pixels.

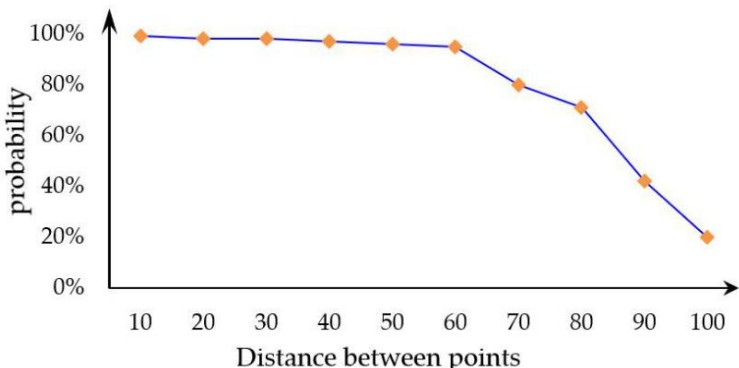

**Figure 15.** Threshold analysis of distance constraint.

### 4.3. Evaluation Index

Accuracy is the measure of the proportion of information that is correctly extracted relative to correct information that is not extracted, and the recall is used to measure how much of the extracted information is correct. Accuracy and recall are essential parameters in the identification of road extraction. The identification of road extraction requires both high precision and high recall rate. Therefore, the F-measure was also introduced to measure it. The formula is as follows:

$$Precision = \frac{TP}{TP + FP} \tag{18}$$

$$Recall = \frac{TP}{TP + FN} \tag{19}$$

$$IoU = \frac{TP}{TP + FP + FN} \tag{20}$$

$$F1 = 2 \times \frac{Precision \times Recall}{Precision + Recall} \tag{21}$$

Here, TP is the length of the correctly extracted roads, FP is the length of nonroad pixels extracted as roads, and FN is the length of roads extracted as nonroads.

In addition, the number of seed points and running time, as an indicator of the degree of automation, was focused on in analysis and research.

### 4.4. Test Set

For accuracy evaluation, considering that rural roads cover a wide area and are characterized by irregular curvature changes, narrow widths, and diverse surface materials, we used different scenes in rural areas. First, the mountain road, as a typical rural road, is limited by the terrain and has the characteristics of narrow width and irregular curvature change. We conducted experimental analyses on this scene and showed the reliability of our different steps. Second, we took a single village as the scene: not only the roads outside the village, but also the roads inside the village and the roads connecting the village and outside should be considered. In this way, the problem of diverse pavement materials can be addressed, and the similarity between roads and farmland can be highlighted. Finally, forest roads are a special type of rural road, and their connection with small towns should also be considered. In addition, to show that the template matching method road extraction results have topological relationships and can be managed, we used different color routes to distinguish different vectors in the first experiment.

### 4.5. Experimental Results and Analysis

#### 4.5.1. Experiment 1

The GF-2 image shown in Figure 16 has a spatial resolution of 0.8 m and a size of 6000 × 6000 pixels and covers a mountainous area in Hubei Province, China (see Table 1 for specific information). Due to the influence of mountain terrain, the ground truth map

shows that the mountain highway obviously presents large local curvature transformations. This is in sharp contrast to the small curvature changes and straightness of urban roads, and, different from the complex road networks of cities, there are many dead- end roads in mountainous areas. As the image covers a large area, we zoomed in and analyzed local road images with obvious features. In Figure 16b, an ordinary rural mountain road is shown, and the curvature transformation is not dramatic. In Figure 16c, the local road direction exhibits a great deflection, and the maximum angle deflection reaches more than 120 degrees. In Figure 16d, not only the characteristics of curvature transformation but also occlusion problems from trees on local roads are present.

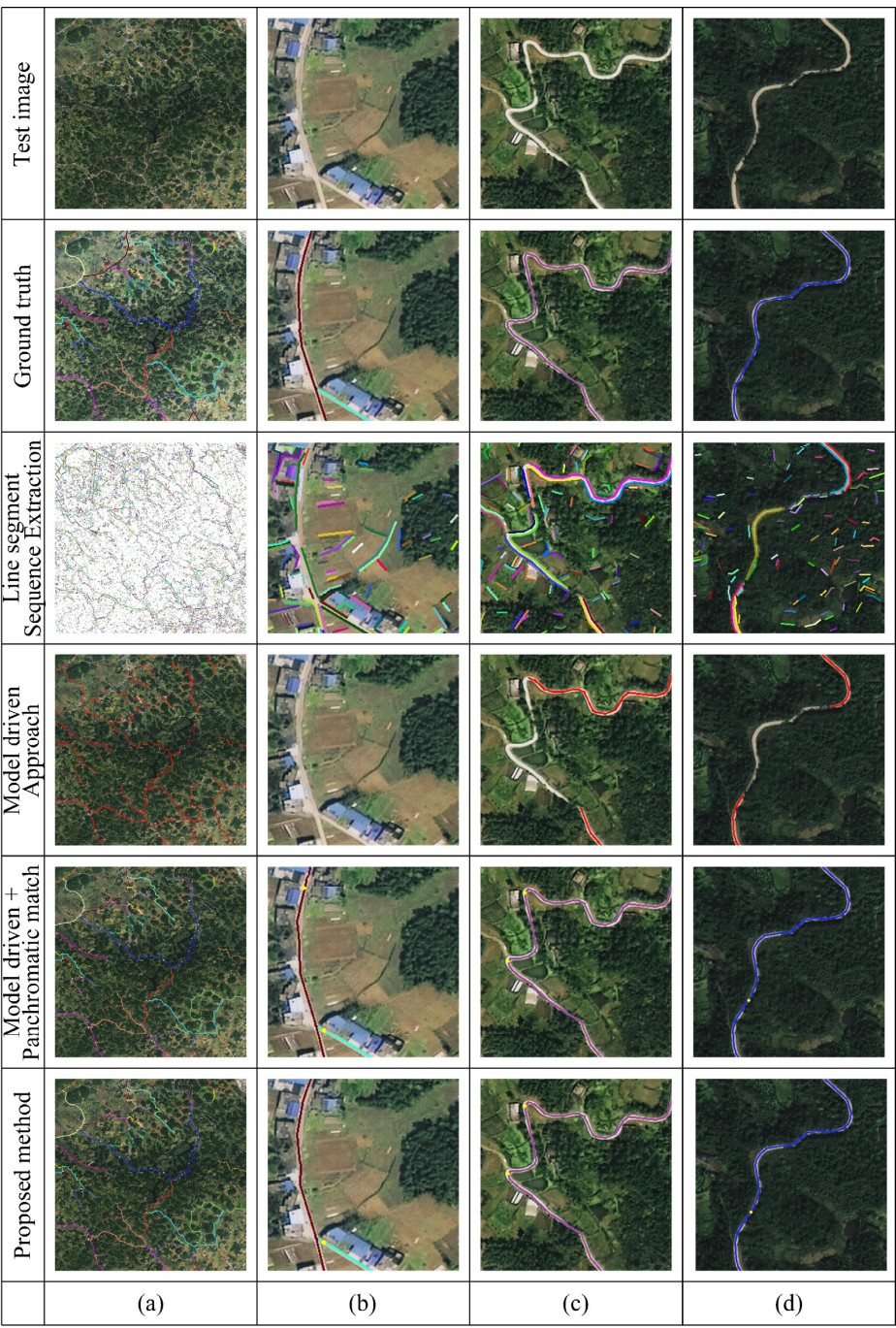

**Figure 16.** This is a figure. Extraction results of different steps from GF-2 image in a rural area (different colored curves represent different routes).

Figure 16b shows that, in the local experimental scene, the road is adjacent to buildings and farmland. Using our line sequence method, some road edges could be extracted continuously, but some road edges could be extracted well. In this case, because the edges on both sides of the road did not meet the length and parallel constraints, the model-driven method did not extract the road at this stage. On this basis, the panchromatic matching model was used to track the road section. In the area where the gray levels of roads and buildings are close to each other, the upper side needed to be supplemented. On the lower side, the seed points were not obtained by using the model-driven method, so they needed to be input manually. After adding the HSV matching model, we could see that the algorithm could distinguish between roads and neighboring houses through the improvement in the recognition of different features in the color space. As a result, the upper seed point no longer needed to be input, which verifies the effectiveness of the HSV spatial matching model.

The scene in Figure 16c includes farmland, trees, and 120° turning roads. It is easy to see that some trees visually block the road. Because of this, there are two faults in the sheltered position of the line sequence. As a result, the model-driven method is unable to extract the road between the fault zones. Similarly, the broken line sequences at the turning points of the roads greater than 120° cannot be extracted directly. Based on the model-driven method, we added two points at the corner of the road. Through the acquisition of sample information, through step growth processing, shadow crossing was completed, and road section extraction was realized.

In Figure 16d, in the scene, the road is located in the middle of the forest area, and the pavement material is concrete pavement. We can see that trees form visual occlusions on the road and form two occluded road areas. Thus, the line sequences between the two occluded road regions were not long enough to meet the length constraint. As a result, the regional model-driven method was unable to directly extract the road, resulting in road fracture. In view of this situation, we added a new seed point here. After a new point is manually input, the line sequence information on both sides of the road can be used. In addition to the difference in the texture spectrum between the road and nonroad areas on the road section, the covered area could be crossed by expanding the step size.

In Table 2, the results of different steps are quantitatively compared and analyzed. Using the model-driven method, the control precision could reach 99.43%, which shows that our strict constraints played a key role in the control accuracy. However, while the model-driven method could control the precision, the integrity was low, only 71.69%. On the basis of the model-driven method, we compared the combination of the model-driven method and panchromatic matching model and the proposed method. There was little difference between the precision, recall, IoU, and F1 of the two methods, which could reach more than 99%. This shows that the sample-driven method can improve the quality of the model-driven method, and the result is suitable for the requirements of actual production. However, regarding the efficiency, our extraction method can decrease the degree of human participation and the operation time, which shows the necessity of adding an HSV spatial matching model.

**Table 2.** Information table for comparison of results of different steps.

|  | Model-Driven Approach | Model-Driven + Panchromatic Match | Proposed Method |
|---|---|---|---|
| Recall (%) | 71.99% | 99.64% | 99.71% |
| Precision (%) | 99.43% | 99.49% | 99.54% |
| IoU (%) | 71.69% | 99.14% | 99.26% |
| F1 (%) | 83.51% | 99.57% | 99.63% |
| Seed Points | 0 | 91 | 83 |
| Time(s) | 136 | 1159 | 1006 |

### 4.5.2. Experiment 2

Figure 17 shows a Pleiades image of a rural island area in Liaoning Province, China, with a spatial resolution of 0.5 m and a size of 5000 × 5000 pixels. First, from the perspective of the full image, limited by economic development, the pavement types include concrete roads, gravel roads, and soil roads, and the corresponding pavement images present different spectral characteristics.

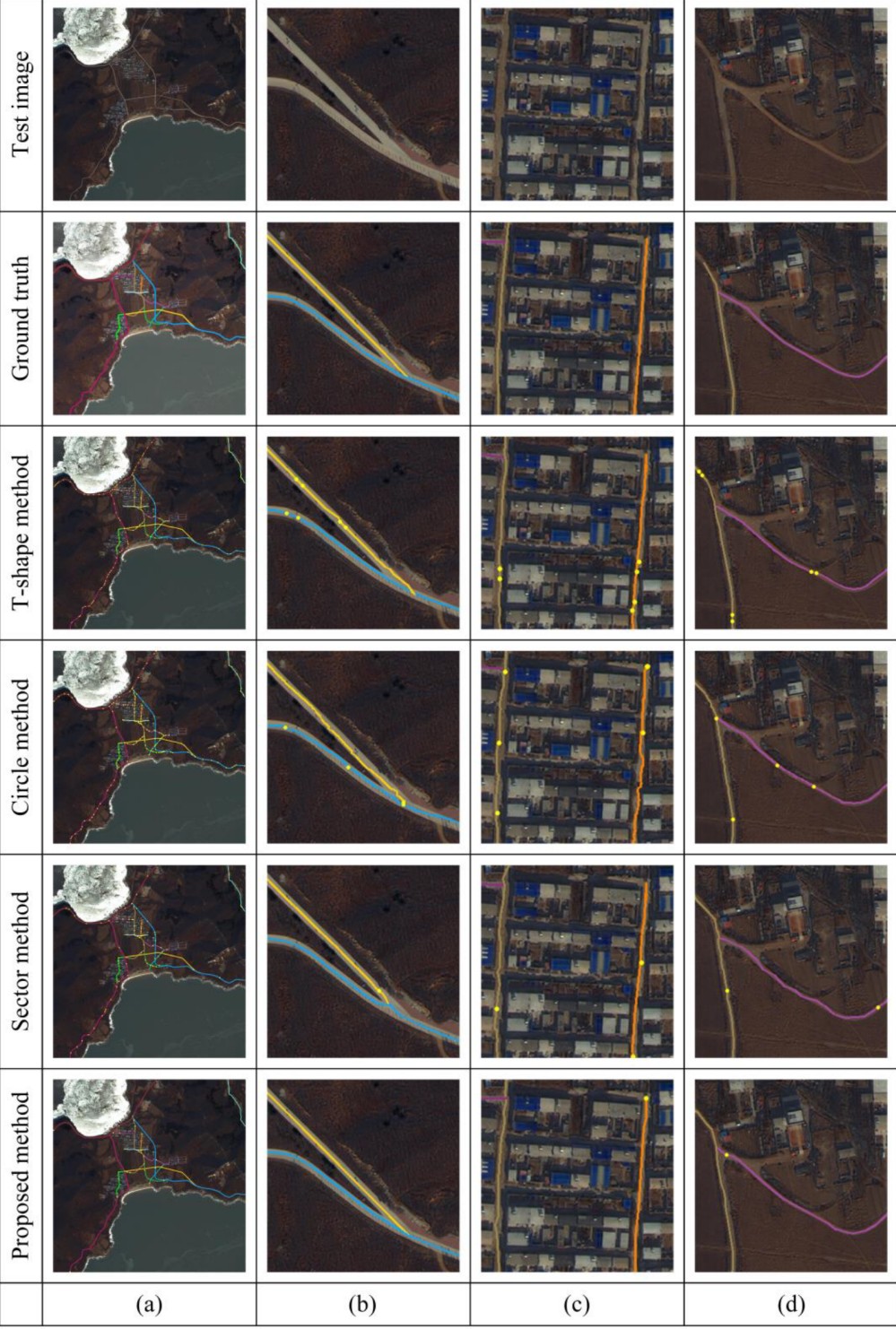

**Figure 17.** This is a figure. The results of road extraction with different algorithms in the rural island area of Pleiades (different colored curves represent different routes).

Figure 17b shows higher grade rural roads. Although there are some tree shadows on both sides of the road, the spectral contrast between the concrete road and the neighborhood is strong. Our algorithm can extract enough length of line sequence information, so it can automatically extract the road section. Due to the tree interference, the circle method [39] and T-shape method [38] had difficulty automatically extracting the road section and needed multiple input points. The sector method [40] adds the MLSOH operator, which can predict the tracking direction by using the constraint of the structure information and can overcome some influence of the tree shadows. However, due to the interference from the trees in the process of line segment extraction, the line segment was broken, so this method required point supplement processing in this section.

In the image in Figure 17c, sand and concrete pavement are present in the local image at the same time. The left side is a gravelly road with less occlusion. Using our method, we can extract longer line sequences. On this basis, sand and gravel roads can be extracted automatically and completely. On the right side of the concrete pavement, our method was disturbed by trees on the upper side, so it needed to be supplemented. Compared with other methods, the T-shaped and circular templates were affected by the houses on both sides in the matching process. Multiple point filling was required. The sector descriptor used the semantic information that the edges of the buildings on both sides were consistent with the direction of the road. This can reduce the influence of adjacent objects on matching and tracking, so the number of complementary points was lower.

The image shown in Figure 17d is a rural dirt road. It is obvious that the color of the road surface tends to be consistent with the farmland on both sides. These conditions strongly interfere in the matching model based on texture features. Therefore, the T-shaped [38] and circular templates [39] needed to be filled with multiple points, while the sector descriptor [40] used the semantic information of the road edge. This can accurately capture the range of road direction transformation, so only two input points needed to be added manually. Our method can connect the line segments of the same edge to form a line sequence. Compared with a single line segment, this can more completely express road edge information. The HSV matching model was used to expand the accuracy of matching. Therefore, our method required an input of only one point.

Table 3 shows a comparison of the results of different methods. The precision, recall, IoU, and F1 of the four methods were similar. The key difference was found for the 14 km road. To analyze the advantages and disadvantages of different algorithms, the corresponding human participation and algorithm running time were compared. In contrast, the T-shaped method [38] and circle method [39] were associated with the greatest manual participation. This shows that the use of only road texture information is readily affected by noise interference. The sector method [40] combines line segment structure information with texture information. The lack of road extraction can greatly reduce the efficiency. Our method uses a model-driven method and a sample-driven method. From automation to semi-automation, the efficiency of the algorithm was greatly improved. Additionally, the accuracy of road extraction results was ensured.

**Table 3.** Comparison information of different methods.

|  | Circle Method | T-Shape Method | Sector Method | Proposed Method |
|---|---|---|---|---|
| Recall (%) | 99.52% | 99.49% | 99.61% | 99.73% |
| Precision (%) | 99.66% | 99.40% | 98.93% | 99.39% |
| IoU (%) | 99.19% | 98.90% | 98.54% | 99.12% |
| F1 (%) | 99.59% | 99.45% | 99.27% | 99.56% |
| Seed Points | 162 | 356 | 78 | 28 |
| Time(s) | 2698 | 4201 | 1358 | 310 |

### 4.5.3. Experiment 3

The GeoEye-1image of Hobart, Australia, shown in Figure 18, has a spatial resolution of 0.41 m and a size of 5000 × 5000 pixels. Figure 16 clearly shows that the town is connected with a forest area. Therefore, asphalt roads and concrete roads are the main pavement types in this area, and some sand and gravel cover the concrete pavement in the forest area. Therefore, the texture homogeneity of the concrete pavement in the forest area is low. Therefore, we selected three groups of different roads in the area for display.

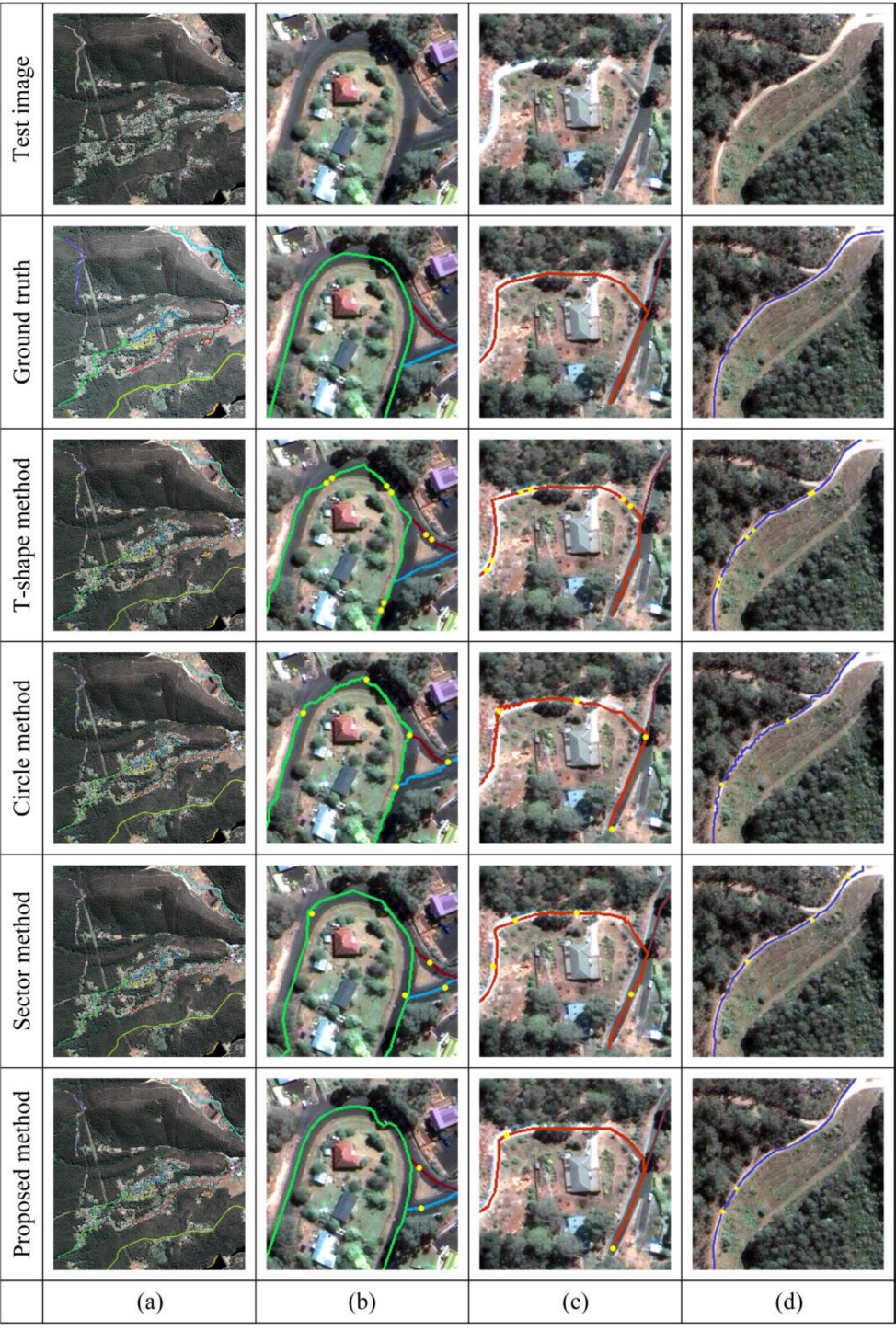

**Figure 18.** This is a figure. The results of road extraction with different algorithms in the town of GeoEye-1 (different colored curves represent different routes).

In Figure 18b, tree occlusion is present at the turning point of the small-town road. Additionally, there are some tree shadows on both sides of other roads. Therefore, the circle method [39] and T-shaped method [38] are limited by this, so it was difficult to automatically extract the road section, and multiple points needed to be input. The sector method [40] with the MLSOH descriptor could complete the extraction of the road section. The method of expanding the step size was used to track the area occluded by trees. In our algorithm, the occluded areas could be connected by a line sequence. Therefore, considering the length of the line segment, the automatic extraction of the road segment could be directly completed through the model-driven method. This shows that the model-driven method has some advantages in overcoming occlusions.

In the urban boundary area, the urban asphalt pavement is connected with the concrete pavement in the forest area, and this connection is located in a tree-sheltered area. Both the T-shaped method and circle method required repair. The sector descriptor could span the partial shadow by increasing the step size. However, additional input points were needed on both sides of the shadow crossing. However, our method obviously had the advantage of fewer input points. Similarly, as shown in Figure 18c, in the area where the trees and roads are connected, the number of input points for our method is the same as that of the circle template [39], two input points. The T-shaped template [38] and sector descriptor [40] needed 5 and 3 input points, respectively, which shows the automation advantage of our algorithm.

In Figure 18, the total mileage of rural roads is 11.4 km. Compared with that in Figure 17, the road surface is covered with more sand and gravel. As a result, more artificial input points needed to be added for our method in this experiment. As shown in Table 4, that road interference was still the key factor hindering the improvement in algorithm automation. Although our method required more seed points in this experiment than the other three methods, it still performed better, which further shows the reliability of our method.

**Table 4.** Comparison information of different methods.

|  | **Circle Method** | **T-Shape Method** | **Sector Method** | **Proposed Method** |
|---|---|---|---|---|
| Recall (%) | 99.42% | 99.37% | 99.44% | 99.47% |
| Precision (%) | 98.19% | 98.73% | 98.36% | 98.82% |
| IoU (%) | 97.63% | 98.12% | 97.82% | 98.31% |
| F1 (%) | 98.80% | 99.05% | 98.90% | 99.15% |
| Seed Points | 142 | 152 | 68 | 54 |
| Time(s) | 2016 | 2648 | 1149 | 722 |

### 4.5.4. Analysis of Experimental Results

In order to evaluate the effectiveness of the algorithm proposed in this paper, the total number of seed points and the total time used in Experiments 2, 3 were statistically analyzed, and the precision, recall, and F1 were analyzed by chi-square test. The chi-square test formula is as follows:

$$\chi^2 = \sum \frac{(A - E)^2}{E} \tag{22}$$

where $A$ is the actual value, $E$ is the expected value, and $E$ is set to 1 in the calculation. $\chi^2$ is the test result. The smaller the $\chi^2$ value is, the closer the actual value is to the expected value.

The main contribution of the proposed algorithm is to improve the degree of automation by reducing the number of seed points and the time used. As shown in Figure 19a,b, the number of seed points and time of the proposed algorithm are much lower than those of the other three comparison algorithms. The number of seed points was reduced by more than 40%, and the time was reduced by more than 55%. Because the four methods are semi-automatic road extraction algorithms with manual participation, the precision, recall, and F1 were all above 98% in the experiment, so the difference of chi-square test results was

not great. However, the algorithm proposed in this paper is more suitable for rural roads with the prominent curvature difference, narrow widths, and diverse pavement materials. Therefore, as shown in Figure 19c, the calculation results of the algorithm proposed in this paper are still better than other methods, and the approximation between the experimental results and the expected results is improved by more than 8%.

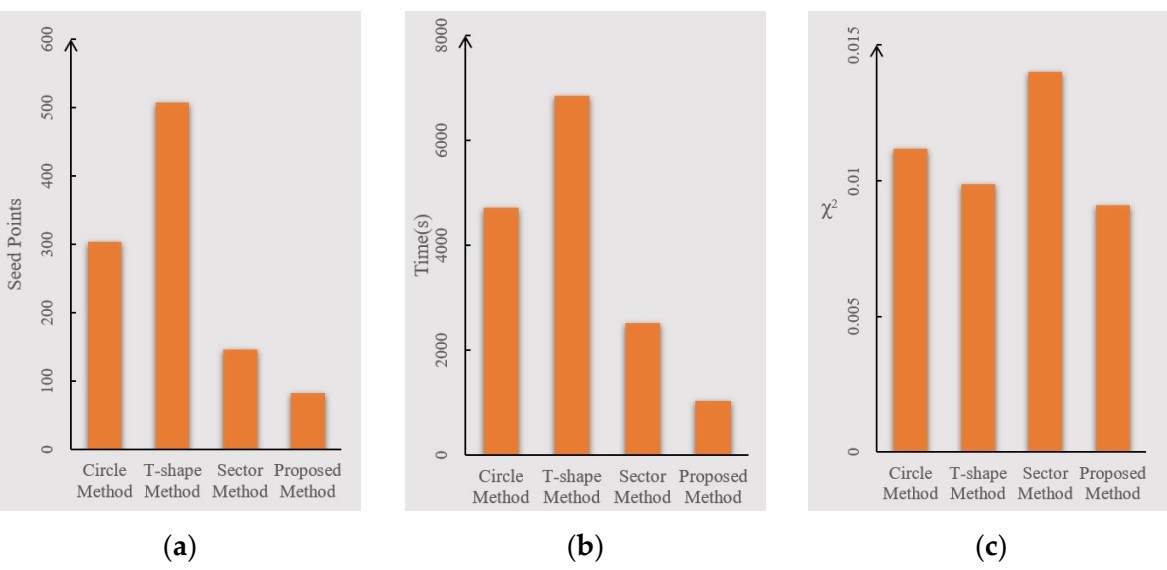

**Figure 19.** Statistical chart of experimental data analysis. (**a**) Statistical chart of total number of seed points; (**b**) statistical chart of total time used; (**c**) chi-square test results of precision, recall, and F1.

## 5. Discussion

We combined a model-driven method with a sample-driven method to maximize the degree of automation in road extraction in rural areas on the basis of ensuring the accuracy. Compared with other methods, the advantages of our method are fully reflected in the following three aspects:

(1) Optimization of the model-driven approach

The model-driven method is completed on the basis of preprocessing. We proposed a line sequence extraction method instead of relying solely on line segments, which greatly improves the ability to fit curved road edges and provides sufficient prior length conditions for the determination of the road edges by the model-driven method. The interference from shadow occlusions and visual occlusions in road extraction can be reduced by parallel constraints and gray mean constraints on the edge line of the line segment and by overcoming the local noise with the overall balance. This is obvious in Figure 18b, when tree occlusion occurs. Since the area covered by trees is obviously different from the road, the problem can be solved if a deep learning model is adopted, but otherwise, the trees are mistaken as roads, thus reducing the overall accuracy of road extraction. By using the connection between different line segments in the line sequence, we can realize road extraction in the covered area through the overall analysis of the uncovered road area on both sides of the adjacent line sequence. Although the integrity of the extraction was reduced, this needs to be carried out in follow-up sample-driven methods.

(2) Improvement in the sample-driven method

Compared with the ordinary sample-driven method, we first modified the MLSOH model by extending the line segment into a line sequence to reduce the interference from structural information in other directions with the road structural information. For example, structural information perpendicular to the road can be weakened for analysis, which is convenient for improving the accuracy of direction prediction. Second, we combined the panchromatic matching model with the HSV spatial matching model to improve the contrast between road and nonroad areas. Especially when roads and nonroads

have similar texture, the accuracy of road matching is further improved by dynamically allocating the weight in HSV space.

(3) Effective connection of the model-driven method and sample-driven method

Due to the prominent differences in the road curvature, paving materials, road width, and other characteristics of rural roads, the model-driven method has the advantage of a high degree of automation. However, the method manifests so many different problems through the theoretical model, which inevitably results in low accuracy and integrity. The sample-driven method can complete road extraction with high precision through the manual input of sample points or the establishment of sample sets. Our approach, a model-driven-to-sample-driven road extraction method, is a combination of the strengthening of the model-driven method to extract the constraints, the improvement of the structure of the sample-driven method, and the use of texture and HSV space in the analyses. On the premise that the accuracy and recall rate of the evaluation indicators reached more than 98%, compared with other methods, the automation of this algorithm increased by more than 40% and greatly improved the automation degree of the algorithm.

## 6. Conclusions and Future Work

To address the low automation degree in rural road extraction caused by prominent curvature differences, complex road paving materials, and narrow road width, we proposed a model-driven-to-sample-driven for rural road extraction method that applies line sequences to curves and connects panchromatic images to HSV space to improve the contrast between road and nonroad areas. In this work, we proposed a line sequence detection method through constraint analysis, seed point formation, and other steps from a model-driven perspective to extract part of the road sections. In this process, we ensured the accuracy of road center point extraction by strictly constraining the road feature information. Then, based on the seed points obtained by model-driven and manual input, the MLSOH model was improved to make the direction prediction results more suitable for the curvature changes of rural roads. With the application of the panchromatic and HSV matching model, the ability of the sample-driven method to overcome noise interference and occlusions was improved. Finally, the experimental verification showed that the proposed method can ensure the quality of rural road extraction and significantly reduce manual participation. However, this method still exhibits the problem of extensive human participation when facing complex roads, which necessitates further development of follow-up work.

**Author Contributions:** Conceptualization, J.D. and R.M.; funding acquisition, J.D.; methodology, J.D.; project data curation, L.G. and Z.S.; resources, J.W.; writing—original draft, J.D. and R.M. All authors have read and agreed to the published version of the manuscript.

**Funding:** This research was funded by The National Natural Science Foundation of China (42071428); The National Natural Science Foundation of China (420713743); Liaoning Provincial Department of Education Project Services Local Project under Grant (LJ2019FL008); Beijing Key Laboratory of Urban Spatial Information Engineering (2020221); Key Laboratory of Surveying and Mapping Science and Geospatial Information Technology of Ministry of Natural Resources (2020-3-5).

**Institutional Review Board Statement:** Not applicable.

**Informed Consent Statement:** Not applicable.

**Data Availability Statement:** Not applicable.

**Acknowledgments:** The authors wish to thank the editors and reviewers.

**Conflicts of Interest:** The authors declare no conflict of interest.

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
