# Peer review of "A Model-Driven-to-Sample-Driven Method for Rural Road Extraction"

_remotesensing, doi:10.3390/rs13081417_

Round 1

Reviewer 1 Report

It is suggested to divide the Introduction paragraph in shorter sub-paragraphs to better orient the summary of state-of-art and background information to the research. An option could be to divide between methods and potentialities that belong to the state-of-art of research.

Author Response

Response to Reviewer 1 Comment

Point 1: It is suggested to divide the Introduction paragraph in shorter sub-paragraphs to better orient the summary of state-of-art and background information to the research. An option could be to divide between methods and potentialities that belong to the state-of-art of research.

Response 1: Thank you very much for your patience in reviewing and giving affirmation and suggestions to this paper. Now this paper has been revised according to your suggestions.

Reviewer 2 Report

The language quality of this manuscript is mostly fine, in particular in the introduction and conclusion. In some parts of the technical chapters the language is clumsier, and there it becomes harder to understand the content. E.g.: the contour grouping described in line 196-208. ‘colors’ can here only refer to the visualization graphics. What is described is an algorithm using labels for groups which are visualized by colors. More flow-chards not only for the overview in Fig. 1 but also for detailed description such ass line grouping would help.

Line 118: ‘… the country’s total highway milage. …’ I know, the American term ‘highway’ is very fuzzy and broad. But to me a three-meter-wide gravel road is definitely not a highway. Why not ‘… the country’s total road milage. …’

Line 256: ’… but in the panchromatic image, the two colors are similar …’ a panchromatic image has only intensities, no colors. Better: ‘… but in the panchromatic image, the two regions share similar intensities …”.

Line 282: In the context of roads, the term ‘texture’ refers to deviations from planarity of the road surface. In the context of image-processing, it refers to repetitive spatial patterns such as in fabrics, or irregular spread similar spatial and intensity repetitions. There are many books and thousands of smarts papers on texture analysis. All that would never fit in a stripe of only 2 pixels width. Formula (4) speaks only of mean intensities. Thus, the term ‘texture’ should be avoided here. There may in fact be texture present, when an old asphalt road is repaired piecewise, using patches of fresh much darker material, or when a gravel road has large chuckholes (filled with water or dirt), or when road marks or grafitti are painted on the surface, or when vehicles, animals or persons are present. But then things can become really difficult!

Line 350-355 contains the text for Fig. 9. It is said that the process of road prolongation is visualized. But in the figure caption it is said that (a) shows a graph. The same term ‘graph’ is used there for (e) which displays a histogram of orientations. Be carful with the term ‘graph’. It is a mathematical object consisting of vertices and edges. A road graph would be a topological representation of a road network and its crossroads. A graph of line segments would represent grouping information, and would be very large for such scene and look very different. I see no graph in 9(a) or (e). 

Line 386-387 mentions ‘texture’ again and then ‘feature’ and ‘convergence’: ‘…; thus, the textures of road-adjacent features in the panchromatic image have convergence characteristics. … ‘ Also here I see no texture analysis, as well no feature-vector of a pattern recognition machine, and no convergence of a mathematical sequence or iterative algorithm. This kind of language triggers the wrong associations. What you probably want to say is, that on- and off-road materials can be very similar. Another detailed flow chard may help to explain the template matching and the conditional branching between intensity and color model.

(9) I don’t like num_i in formulae. Better use |circ_i|.

Page 12-13: My impression is that you are not really using ‘template matching’ in the sense of cross correlation between pixel matrices. Instead histograms are obtained from the patches and used as features for matching. So also the term ‘template’ triggers wrong associations.  

Starting from page 10 the PDF contains Chinese characters. Probably these are references or hints on missing references. This should not happen and shows a certain lack of diligence in the authors. Before I submit a PDF to a journal I check once more every page for such annoying little mistakes.

The paper does not present a revolutionary new idea or method. Instead, it proposes are carefully assembled arrangement of known and proven methods for road extraction from satellite imagery. Focus is on tiny rural roads, where fully automatic recognition frequently fails. I particularly like the very practical touch of the paper. It has become modern to propose so-called AI methods for such recognition problems, i.e., deep-learning convolutional perceptrons. The paper sees these learning machines in the category of sample-driven methods. As contrast to this, the classical existing methods, that explicitly search for road edges, and then follow the road inspecting road- and non-road colors and textures sequentially along the way in good continuation, are categorized as model-driven methods. The latter are seen to have many advantages, including an output format that meets the specifications for geographic data. Model-driven methods – which are pejoratively called hand-made-feature-based today – have a proven robustness and precision, and they are diligently engineered, giving the tools for failure explanation and debugging, as well as insight where what knowledge is utilized. The paper gives sufficient reference to up-to-date literature on this issue.

The week point in model-based approaches lies in the parameters (stripe-widths, thresholds, etc.). Often their values are chosen rather heuristically based on very limited empirical evidence, or without awareness of the corresponding problems of knowledge utilization. I think in this paper most parameter choices are reasonably justified. There is a diligent analysis for some thresholds around page 15 and 16.  Also the language is more understandable in these chapters. However, there are many histograms used here. And I would like the bin sizes to be discussed in more details – why 16 intensity bins, 20 degrees per bin, etc?

The empirical part of the work uses imagery from three different satellite systems depicting regions in Australia as well as Hubei province of China. Comparison is with other state-of-the-art model-based methods. It is claimed that the perceptrons reach no more than 95%, and the output format does not fit. Still, a little more work here would make the paper more convincing. Why not write a quick little program translating the output formats properly and give some number for a standard perceptron as well? The results presented on pages 17-23 are impressing. I like this work!

Author Response

Response to Reviewer 12 Comments

Point 1: The language quality of this manuscript is mostly fine, in particular in the introduction and conclusion. In some parts of the technical chapters the language is clumsier, and there it becomes harder to understand the content. E.g.: the contour grouping described in line 196-208. ‘colors’ can here only refer to the visualization graphics. What is described is an algorithm using labels for groups which are visualized by colors. More flow-charts not only for the overview in Fig1 but also for detailed description such assembly line grouping would help.

Response 1:Thank you very much for your suggestion. I'm sorry for the misunderstanding. The ‘colors’ mentioned here is only used to show different information extraction results. According to your suggestion, this paper rewrites this paragraph according to the result of different steps of line segment sequence extraction in Figure 2, which can effectively express the flow of the algorithm. The details are as follows: “Due to the continuity of Canny edge tracking results, the edge points with 8-connected relations are grouped using the edge points extracted by Canny, and the same grouped edge as one connected component. As shown in Figure 2 (b), the same edge group is given the same color for display. On this basis, line segments are extracted by using the Dai’s method . Since the results of line segments are independent, in order to clearly express the differences between line segments, as shown in Figure 2 (c), we use color to distinguish line segments, and assign different colors to different line segments for display. Finally, the line segments in the same group are processed to form different line segment sequences.” Then this article also makes some changes in the wording of other chapters. Thank you for your patience.

Point 2: Line 118: ‘… the country’s total highway milage. …’ I know, the American term ‘highway’ is very fuzzy and broad. But to me a three-meter-wide gravel road is definitely not a highway. Why not ‘… the country’s total road milage. …’

Response 2: Thank you very much for your suggestion. The meaning of this article is really "Occupying the total mileage of the national road...". Therefore modify based on your suggestions, change to'...the country's total road milage....".

Point 3: Line 256:’… but in the panchromatic image, the two colors are similar …’ a panchromatic image has only intensities, no colors. Better: ‘… but in the panchromatic image, the two regions share similar intensities …”.

Response 3: Thank you very much for your suggestion. According to your suggestion, we have amended it. Now, "… but in the panchromatic image, the two colors are similar …" is changed to "… but in the panchromatic image, the two regions share similar intensities …".

Point 4: Line 282: In the context of roads, the term ‘texture’ refers to deviations from planarity of the road surface. In the context of image-processing, it refers to repetitive spatial patterns such as in fabrics, or irregular spread similar spatial and intensity repetitions. There are many books and thousands of smarts papers on texture analysis. All that would never fit in a stripe of only 2 pixels width. Formula (4) speaks only of mean intensities. Thus, the term ‘texture’ should be avoided here. There may in fact be texture present, when an old asphalt road is repaired piecewise, using patches of fresh much darker material, or when a gravel road has large chuckholes (filled with water or dirt), or when road marks or grafitti are painted on the surface, or when vehicles, animals or persons are present. But then things can become really difficult!

Response 4: Thank you very much for your suggestion. According to your suggestion, we have changed the ‘texture’ to ‘gray mean’.

Point 5: Line 350-355 contains the text for Fig. 9. It is said that the process of road prolongation is visualized. But in the figure caption it is said that (a) shows a graph. The same term ‘graph’ is used there for (e) which displays a histogram of orientations. Be carful with the term ‘graph’. It is a mathematical object consisting of vertices and edges. A road graph would be a topological representation of a road network and its crossroads. A graph of line segments would represent grouping information, and would be very large for such scene and look very different. I see no graph in 9(a) or (e).

Response 5: Thank you very much for your suggestion. Due to improper wording of this article, you have been puzzled. According to your suggestion, it has been amended. Now, we have changed (a) "graph" to "image". (e) "graph" has been changed to "results".

Point 6: Line 386-387 mentions ‘texture’ again and then ‘feature’ and ‘convergence’: ‘…; thus, the textures of road-adjacent features in the panchromatic image have convergence characteristics. … ‘Also here I see no texture analysis, as well no feature-vector of a pattern recognition machine, and no convergence of a mathematical sequence or iterative algorithm. This kind of language triggers the wrong associations. What you probably want to say is, that on- and off-road materials can be very similar. Another detailed flow chard may help to explain the template matching and the conditional branching between intensity and color model.

Response 6: Thank you very much for your suggestion. Due to improper wording of this article, you have been puzzled. According to your suggestion, it has been amended. Now, we have changed ‘…; thus, the textures of road-adjacent features in the panchromatic image have convergence characteristics. …’to ‘the spectral characteristic of adjacent ground objects in the panchromatic image are similar to those of roads’.

Point 7:(9) I don’t like num_i in formulae. Better use |circ_i|.

Response 7: Thank you very much for your suggestion. According to your suggestion, we have changed the ‘numi’ to ‘circi’.

Point 8: Page 12-13: My impression is that you are not really using ‘template matching’ in the sense of cross correlation between pixel matrices. Instead histograms are obtained from the patches and used as features for matching. So also the term ‘template’ triggers wrong associations.

Response 8: I'm sorry for the confusion caused by the unclear statement. ‘Template’ mentioned here specifically refer to the construction of a circular template area in which the pixel values are counted. They are only used to determine the area and are not used to match between the template and the template.

Point 9: Starting from page 10 the PDF contains Chinese characters. Probably these are references or hints on missing references. This should not happen and shows a certain lack of diligence in the authors. Before I submit a PDF to a journal I check once more every page for such annoying little mistakes.

Response 9: I am very sorry that the Chinese appeared in PDF due to personal negligence. All these have been modified.

Point 10: The paper does not present a revolutionary new idea or method. Instead, it proposes are carefully assembled arrangement of known and proven methods for road extraction from satellite imagery. Focus is on tiny rural roads, where fully automatic recognition frequently fails. I particularly like the very practical touch of the paper. It has become modern to propose so-called AI methods for such recognition problems, i.e., deep-learning convolutional perceptrons. The paper sees these learning machines in the category of sample-driven methods. As contrast to this, the classical existing methods, that explicitly search for road edges, and then follow the road inspecting road- and non-road colors and textures sequentially along the way in good continuation, are categorized as model-driven methods. The latter are seen to have many advantages, including an output format that meets the specifications for geographic data. Model-driven methods – which are pejoratively called hand-made-feature-based today – have a proven robustness and precision, and they are diligently engineered, giving the tools for failure explanation and debugging, as well as insight where what knowledge is utilized. The paper gives sufficient reference to up-to-date literature on this issue.

Response 10: Thank the reviewer for your patience in reviewing the paper and your affirmation.

Point 11: The week point in model-based approaches lies in the parameters (stripe-widths, thresholds, etc.). Often their values are chosen rather heuristically based on very limited empirical evidence, or without awareness of the corresponding problems of knowledge utilization. I think in this paper most parameter choices are reasonably justified. There is a diligent analysis for some thresholds around page 15 and 16.  Also the language is more understandable in these chapters. However, there are many histograms used here. And I would like the bin sizes to be discussed in more details – why 16 intensity bins, 20 degrees per bin, etc?

Response 11: Thanks to the reviewers for their patience in reviewing the paper and giving suggestions and affirmations. Firstly, this paper refers to the document [1] to determine the histogram unit when predicting the angle. The document mainly focuses on urban roads with small curvature changes and sets the angle unit to 10°. Since this article is aimed at rural road extraction, in order to adapt to the large curvature of rural roads, this article sets the histogram unit established for local road direction prediction to 20°. Then, when building the color histogram, according to the literature [2], the paper requantized the gray value and linearly compressed the 8 Bit image to 4 Bit to solve the problem of poor gray level homogenization of the road image. Accordingly, this paper divides the gray scale into 16 levels.

[1] Dai, J.; Zhu, T.; Zhang, Y.; et al. Lane-level road extraction from high-resolution optical satellite images. Remote Sens. 2019,11(22), 2672.

[2] Dai, J.; Du, Y.; Fang X.; et al. Road extraction method for high resolution optical remote sensing images with multiple feature constraints [J]. Int. J. Remote Sens. 2018, 22(005):777-791.

Point 12: The empirical part of the work uses imagery from three different satellite systems depicting regions in Australia as well as Hubei province of China. Comparison is with other state-of-the-art model-based methods. It is claimed that the perceptrons reach no more than 95%, and the output format does not fit. Still, a little more work here would make the paper more convincing. Why not write a quick little program translating the output formats properly and give some number for a standard perceptron as well? The results presented on pages 17-23 are impressing. I like this work!

Response 12: Thank the reviewers for their patience in reviewing the paper and their suggestions and affirmation. According to the actual production demand, the precision of road extraction needs to reach more than 95%, and vectorization data is needed. However, according to the literature, the deep learning road extraction results are usually less than 85%, which has not yet met the production demand, and manual post-processing is needed to complete the vectorization of extraction results. Because our research mainly focuses on the traditional road extraction methods, you mentioned the write a quick little program. At present, due to the time limit, this modification is required to be completed within five days. In addition, we don't know much about this aspect, so it can't be implemented in a short time. I'm very sorry.

Reviewer 3 Report

I read the manuscript and I see that the authors should address the following comments and resubmit the revised version for consideration.

  • The Abstract is vague and very long. It should be reduced to be no more than 250 words, and it should contain how much the task is improved in terms of percentage?
  • A new section should be created entitled "Literature Review" or "Related Work"
  • In the introduction, you have to tell what is need for this research work? What you can get from this? What is the motivation behind this research work? Also, the importance of this research must be added as a paragraph in the introduction section.
  • In the Literature Review section, the authors should mention if there is any research work, which did this earlier? The authors should classify the related work into groups based on the technique used including the methods that are built on a deep learning approach. They can review and include the recent and current related work. They also should describe the limitations of these researches. Moreover, at the end of the Literature Review section, the authors should summarize their findings by indicating gaps or shortcomings in all the reviewed papers. After that, the authors should indicate how their paper would address any or all of the gaps.
  • Please rewrite contributions as points, it is not describing the paper in the current form.
  • The authors wrote “Deep learning models require a large number of prior sample sets, which is prohibitively expensive and laborious. In addition, the road extraction results of deep learning methods lack topological network information, and the results still need extensive intervention before the data can be stored in a database.”; “The existing extraction methods face a contradiction between automation and quality”; Please give references for this claim. There many sentences that need to be referenced.
  • Experimental Data needs more description for example the number of instances used for evaluation.
  • “Prediction of local road direction” means to predict the local road direction based on some features. What are the methods used for prediction? Are these methods machine learning methods? Please clarify. Or, you can replace it with estimating the local road direction if it is suitable.
  • In Table 3 comparison information of different methods, the Circle Method, T-shape Method, Sector Method are proposed by some related works. Please give the references.
  • A comparison with different methods in recent and current related work is required.
  • The authors should conduct other experiments using some other metrics such as precision, recall, F-score, and IoU score to show the stability of the method.
  • For the results of the methods in Tables 2 and 3 against the proposed method, the authors should analyze whether the improvements are significant or not. I suggest performing a statistical analysis test and then according to the p-value, the authors can prove the significance of the improvements.

Author Response

Response to Reviewer 12 Comments

Point 1: The Abstract is vague and very long. It should be reduced to be no more than 250 words, and it should contain how much the task is improved in terms of percentage?

Response 1: Thank you for pointing out this problem. We reduce the abstract part of the paper and increase the percentage of the proposed algorithm advantage. At lines 26-27, add "On the premise that the accuracy and recall rate of the evaluation indicators reach more than 98%, compared with other methods, the automation of this algorithm has increased by more than 40%.”

Point 2: A new section should be created entitled "Literature Review" or "Related Work"

Response 2: Thank you for pointing out this case. I'm very sorry for the trouble caused by the arrangement of this chapter. The structure of this paper is strictly in accordance with the template structure provided by remote sensing, as shown in Figure 1 below. In recent papers on remote sensing, we have not seen writing "Literature Review" or "Related Work" separately, so this part is not added in this paper. The "Related work" part of this paper is reflected in 1. Introduction, including the previous research work on the existing road extraction algorithm, such as lines 42-109, and the analysis of the current situation, types and characteristics of rural roads, such as lines 110-120.

Fig 1. Template structure provided by Remote sensing

Point 3: In the introduction, you have to tell what is need for this research work? What you can get from this? What is the motivation behind this research work? Also, the importance of this research must be added as a paragraph in the introduction section.

Response 3: Thank you for pointing out this issue. As we may not have a deep understanding of your question, we will answer it from the following four aspects.

(1) Research needs: rural roads play a key role in the development of rural regional economy. The informatization of rural roads is the inevitable requirement of the development of the times and the only way of modernization. Compared with urban roads, rural roads occupy a higher proportion in mileage. Taking China as an example, the number of rural roads is 4.2 million km, accounting for 83.8% of the total mileage of the country. In order to speed up the development of rural regional economy, the vectorized data storage of rural roads has become an important task.

(2) The purpose and current situation of the research are as follows: Compared with urban road, rural road has the characteristics of irregular curvature change, narrow road width and diversified pavement materials. The common traditional algorithms used in rural road have poor robustness and low degree of automation; The results of deep learning algorithm are difficult to vectorize, and can’t directly meet the production requirements of data storage.

(3) Why this study was conducted: Due to the fact that a practical method for rural road engineering has not yet been found, it is still necessary to consume a lot of manpower, material and financial resources at present. Therefore, combined with the characteristics of rural roads, in order to improve the degree of automation and accuracy of rural road extraction, according to the actual needs of production, this paper carried out the research of rural road extraction method.

 (4) The importance of the proposed method is as follows: The algorithm proposed in this paper improves the degree of automation of rural road extraction on the premise that the precision, recall rate and other parameters reach more than 98%, and the results have the advantage of vectorization storage, which meets the requirements of engineering practicality.

According to your suggestion, we have revised the paper. such as lines 110-120.

Point 4: In the Literature Review section, the authors should mention if there is any research work, which did this earlier? The authors should classify the related work into groups based on the technique used including the methods that are built on a deep learning approach. They can review and include the recent and current related work. They also should describe the limitations of these researches. Moreover, at the end of the Literature Review section, the authors should summarize their findings by indicating gaps or shortcomings in all the reviewed papers. After that, the authors should indicate how their paper would address any or all of the gaps.

Response 4: Thank you for pointing out this issue.

(1) Due to the structural requirements of the template, this paper does not write "literature review" separately. This paper divides the common and classic road extraction methods into model driven methods and sample driven methods according to whether they need to provide prior samples, which are listed in the introduction.

(2) According to the difference of whether prior samples are needed or not, road extraction methods are divided into model-driven methods and sample-driven methods in this paper. According to your suggestion, this paper first rewrites the model-driven methods according to the geometric features and texture features; secondly, this paper divides the sample-driven methods into deep learning and template matching methods, and rewrites the deep learning methods according to the time sequence. Rewrite as follows: "Mnih and Hinton [1] proposed a road extraction method based on restricted Boltzmann machine (RBM) for the first time by using deep learning technology. 2015, Long et al. [2] proposed the fully convolutional network (FCN), which is the most commonly used road extraction architecture, but needs well annotated samples to train these deep learning models. Since then, various FCN-like architectures have been proposed, including U-Net [3], SegNet [4] and DeepLabV3+ [5]. In this direction, Panboonyuen et al. [6] improved the road extraction accuracy by using SegNet network combined with ELU (exponential linear unit) function, and LMs (landscape metrics) is used to further reduce the misclassification, and finally use conditional random fields (CRFs) to sharpen the extracted roads. The proposed method improved the integrity of road extraction results. However, most of these methods are encoder-decoder structure, in the part of decoder, the boundary accuracy will be reduced, resulting in the discontinuity of road extraction results. In order to solve the problem of discontinuous extraction results, Gao et al. [7] proposed a post-processing method for roads with broken connections. However, given the changeable nature of image condition, the post-processing operations are complex, which reduces the automaticity of road extraction. Zhou et al. [8] proposed a road extraction network based on boundary and topology perception. "

(3) In this paper, at the end of each type of road extraction method, the existing shortcomings of this type of method are stated.

Such as lines 57-64, “However, the method of segmentation and acquisition of image objects does not con-sider the high-level features of the image, such as morphological information and contextual semantic information, and includes pixel aggregation based on spectral features, without making full use of other features of high-resolution remote sensing im-ages [9,10]. Therefore, the object unit obtained by the segmentation method often does not match the shape of the actual target feature, and as a result, the processing results of the object-oriented method cannot be converted into results with actual geographical significance [11,12].”is disadvantages of the segmentation method.

Such as lines 70-74, “In general, although the model-driven method has a high degree of automation, its ex-traction accuracy and completeness are still unsatisfactory in the face of a complex road environment, and due to the lack of human intervention, the vectorization of the results requires extensive intervention.” is disadvantages of the model-driven method.

Such as lines 101-108,“However, the integrity of deep learning road extraction results is usually less than 85%, and the road extraction results are only two or more classification problems. There are no vector topological relationships between the classification results, and the data cannot be directly put into a database. Much manual postprocessing is needed; other-wise, the actual application requirements cannot be met [13,14]. In addition, when there are differences between the training sample sets and test sets, it is difficult for deep learning to realize transfer learning, which means different sample sets need to be selected for road areas with very different characteristics [15,16].”is disadvantages of the deep learning road extraction method.

Such as lines 122-124, “However, with shortcomings in the remote sensing images of complex scenes, the template matching method requires extensive manual intervention.” is disadvantages of the template matching method.

(4) At the end of the introduction, the advantages of the two types of methods are combined, and the algorithm in this paper is proposed. As 136-140 line:“Therefore, on the basis of the template matching method, to improve the degree of automation and precision of rural road extraction, we seek to combine the advantages of the high degree of automation of a model-driven method and the high precision of a sample-driven method and propose a model-driven to sample-driven road extraction method for rural road extraction.”In order to summarize the advantages of the two types of algorithms and combine them, the algorithm in this paper is proposed.

(5) According to your suggestion, this article is revised at lines 142-169, added “This solves the problem that the template matching is low in automation and requires a lot of manual participation.” and added “By improving the multi-scale line segment orientation histogram (MLSOH) model and using a full-color and hue, saturation, value (HSV) spatial interactive matching model to improve the adaptability of traditional template matching methods.”

  1. Mnih, V.; Hinton, G.E. Learning to detect roads in high-resolution aerial images. In Proceedings of the Computer Vision—ECCV, European Conference on Computer Vision, Heraklion, Crete, Greece, 5–11 September 2018, 6316, 210–223.
  2. Long, J.; Shelhamer, E.; Darrell, T. Fully convolutional networks for semantic segmentation. IEEE Conference on Computer Vision and Pattern Recognition. 2015.
  3. Ronneberger, O.; Fischer, P.; Brox, T. U-Net: Convolutional networks for biomedical image segmentation. Medical Image Computing and Computer-Assisted Intervention. 2015, 234–241.
  4. Badrinarayanan, V.; Kendall, A.; Cipolla, R. SegNet: A deep convolutional encoder-decoder architecture for image segmentation. IEEE Trans. Pattern Anal. Mach. Intell. 2017, 39(12), 2481–2495.
  5. Chen, L.; Zhu, Y.; Papandreou, G.; et al. Encoder decoder with atrous separable convolution for semantic image segmentation. Lecture Notes in Computer Science. 2018, 833–851.
  6. Panboonyuen, T.; Jitkajornwanich, K.; Lawawirojwong, S.; et al. Road Segmentation of remotely-sensed images using deep convolutional neural networks with landscape metrics and conditional random fields. Remote Sens. 2017, 9, 680.
  7. Gao, L.; Song, W.; Dai, J.; Chen, Y. Road extraction from high-resolution remote sensing imagery using refined deep residual convolutional neural network. Remote Sens. 2019, 11, 552.
  8. Zhou, M.; Sui, H.; Chen, S.; et al. BT-RoadNet: A boundary and topologically-aware neural network for road extraction from high-resolution remote sensing imagery. ISPRS J. Photogramm. Remote Sens. 2020, 168, 288-306.
  9. Zhang, X.; Wang, Q.; Chen, G.Z.; et al. An object-based supervised classification framework for very-high-resolution remote sensing images using convolutional neural networks. Remote Sens. Lett. 2018, 9, 373-382.
  10. Myints, W.; Gobe, R.P.; Brazel, A.; et al. Per-pixel vs. object-based classification of urban land cover extraction using high spatial resolution imagery. Remote Sens Environ. 2011, 115, 1145-1161.
  11. Liu, W.; Wu, Z.; Luo, J.; et al. A divided and stratified extraction method of high-resolution remote sensing information for cropland in hilly and mountainous areas based on deep learning. Acta Geod. Cartogr. 2021, 50, 105-116.
  12. Zhang, Y.; Zhang, Z.; Gong, J.Y. Generalized photogrammetry of spaceborne, airborne and terrestrial multi-source remote sensing datasets. Acta Geod. Cartogr. 2021, 50, 1-11.
  13. Ren, S.; He, K.M.; Girshick, R.; et al. Faster R-CNN: towards real-time object detection with region proposal networks. IEEE T Pattern Anal. 2017, 39(6), 1137-1149.
  14. Teerapong, P.; Kulsawasd, J.; Siam, L.; et al. Road segmentation of remotely-sensed images using deep convolutional neural networks with landscape metrics and conditional random fields. Remote Sens. 2017, 9, 680–698.
  15. Wang, S.; Mu, X.; He, H. Feature-representation-transfer based road extraction method for cross-domain aerial images. Acta Geod. Cartogr. 2020, 49(5), 611-621.
  16. Xu, Y.; Xie, Z.; Feng, Y.; Chen, Z. Road extraction from high-resolution remote sensing imagery using deep learning. Remote Sens. 2018, 10, 1461.

Point 5: Please rewrite contributions as points, it is not describing the paper in the current form.

Response 5: Thank you for reminding of this point. In fact, the “1. Introduction” of this paper is the contribution of this paper, mainly reflected in:

  1. The results can be vectorized into a library. Deep learning models require a large number of prior sample sets, which is prohibitively expensive and laborious. In addition, the road extraction results of deep learning methods lack topological network information, and the results still need extensive intervention before the data can be stored in a database. In this study, we first develop a model-driven method to automatically extract some rural roads. Second, we adopt a method of human-computer interaction to complete road extraction. The extracted road results include topological information, which meets the requirements of practical engineering.
  2. High degree of automation. The existing extraction methods face a contradiction between automation and quality; that is, the higher the degree of automation, the more difficult it is to control the quality of road extraction; in contrast, the lower the degree of automation, the more difficult it is to guarantee the accuracy. We fully exploit the high precision of the template matching method and improve the template matching method to reduce manual participation. Additionally, on the basis of ensuring accuracy and integrity, a model-driven method with a high strength constraint is introduced to make the algorithm more automatic, giving our proposed method high precision and high recall. This solves the problem that the template matching is low in automation and requires a lot of manual participation.
  3. High compatibility. We fully consider that rural roads exhibit irregular curvature changes, narrow widths and diverse pavement materials in the algorithm design process. By improving the multi-scale line segment orientation histogram (MLSOH) model and using a full-color and hue, saturation, value (HSV) spatial interactive matching model to improve the adaptability of traditional template matching methods. We demonstrate this feature in the actual scene experiment. Mountain roads and forest roads are used to represent irregular curvature transformations and narrow widths. Additionally, different road types, such as roads at junctions between small towns and forest areas, roads in rural villages and roads between farmland, represent the diverse characteristics of road surface materials.

Point 6: The authors wrote “Deep learning models require a large number of prior sample sets, which is prohibitively expensive and laborious. In addition, the road extraction results of deep learning methods lack topological network information, and the results still need extensive intervention before the data can be stored in a database.”; “The existing extraction methods face a contradiction between automation and quality”; Please give references for this claim. There many sentences that need to be referenced.

Response 6: “Deep learning models require a large number of prior sample sets, which is prohibitively expensive and laborious. “Add references [1][2]ï¼› “In addition, the road extraction results of deep learning methods lack topological network information, and the results still need extensive intervention before the data can be stored in a database.”Add references [3][4]ï¼›“The existing extraction methods face a contradiction between automation and quality”Add references [5][6].

[1] Liu, B.; Guo, W.; Chen, X.; et al. Morphological attribute profile cube and deep random forest for small sample classification of hyperspectral image. IEEE Access. 2020, 99, 1-1.

[2] Shao, Z.; Zhou, Z.; Huang, X.; et al. MRENet: Simultaneous extraction of road surface and road centerline in complex urban scenes from very high-resolution images. Remote Sens. 2021, 13(2), 239-239.

[3] Zhang, Y.; Xiong, Z.; Zang, Y.; et al. Topology-Aware Road Network Extraction via Multi-Supervised Generative Adversarial Networks. Remote Sens. 2019, 11(9), 1017.

[4] Ren, Y.; Yu, Y.; Guan, H. DA-CapsUNet: A Dual-Attention Capsule U-Net for road extraction from remote sensing imagery. Remote Sens. 2020, 12(18), 2866.

[5] Zhou, M.; Sui, H.; Chen, S.; et al. BT-RoadNet: A boundary and topologically-aware neural network for road extraction from high-resolution remote sensing imagery. ISPRS J. Photogramm. Remote Sens. 2020, 168, 288-306.

[6] Cao, C.; Sun, Y. Automatic Road Centerline Extraction from Imagery Using Road GPS Data. Remote Sens. 2014, 6(9), 9014-9033.

Point 7: Experimental Data needs more description for example the number of instances used for evaluation.

Response 7: Thank you for your suggestion and I'm sorry for the trouble. Our original intention is for the cohesion of the paper, and the paper needs to meet the requirements of Remote sensing journal reference template. Therefore, we divide the experimental data writing into two parts. One part is in 2.1 experimental data, which briefly introduces the selected image type, production time, technical specifications, and selection area; The other part introduces the road types contained in the image in 3.4 test set; In order to facilitate the experimental analysis, but also to avoid repeated description, in 3.5 experimental results and analysis, this paper introduces the spatial resolution, size and road kilometers of each image.

Point 8: “Prediction of local road direction” means to predict the local road direction based on some features. What are the methods used for prediction? Are these methods machine learning methods? Please clarify. Or, you can replace it with estimating the local road direction if it is suitable.

Response 8: I'm very sorry for the trouble caused to you due to the lack of clarity in this paper.

(1) "Prediction of local road direction" refers to the MLSOH model in our previous paper [1]. In this paper, we improve the characteristics of rural roads and the extraction results of line segments.

(2) This model is a feature analysis method in the field of computer vision, not a classical machine learning method.

(3) In the process of tracking, MLSOH model adopts the way of local region growth. At present, there are few methods in this area, and there is no suitable road direction prediction method.

[1] Dai, J.; Zhu, T.; Zhang, Y.; et al. Lane-level road extraction from high-resolution optical satellite images. Remote Sens. 2019,11(22), 2672.

Point 9: In Table 3 comparison information of different methods, the Circle Method, T-shape Method, Sector Method are proposed by some related works. Please give the references.

Response 9: Thank you for raising this question. The different experimental methods used in Table 3 are introduced in 3.1 comparison methods.

Point 10: A comparison with different methods in recent and current related work is required.

Response 10: I'm sorry for the trouble caused by the writing problem. At present, deep learning road extraction method is a research hotspot, but because we mainly use semi-automatic way to extract road. Through the way of manual input seed points, the road extraction is completed, and the extraction results in the process of manual participation not only ensure the extraction accuracy, the results such as precision, recall, etc. are more than 95%, but also directly realize the vectorization storage. According to this requirement, instead of deep learning road extraction, we choose a classic and two latest semi-automatic extraction methods. Among them, the T-shaped method[1] was published in the Journal of Wuhan University in 2009; the circular method[2] was proposed in 2018 and published in Acta Geodaetica et Cartographica Sinca; the sector method[3] was proposed in 2019 and published in remote sensing. The former is a very classic road extraction algorithm, while the latter two have a certain comparative significance both in terms of time comparison and the quality of published journals.

[1] Lin, X.; Liu, Z. Semi-automatic extraction of ribbon roads from high resolution remotely sensed imagery by T-shaped template matching. Geom. Inf. Sci. Wuhan Univ. 2009, 7147, 293–296.

[2] Lian, R.; Wang, W.; Li, J. Road extraction from high-resolution remote sensing images based on adaptive circular template and saliency map. Acta Geod. Cartogr. Sin. 2018, 47, 950–958.

[3] Dai, J.; Zhu, T.; Zhang, Y.; et al. Lane-level road extraction from high-resolution optical satellite images. Remote Sens. 2019,11(22), 2672.

Point 11: The authors should conduct other experiments using some other metrics such as precision, recall, F-score, and IoU score to show the stability of the method.

Response 11:Thank you very much for your suggestions. The original quality evaluation method proposed by Wiedemann is the general evaluation factor of road extraction. However, according to your suggestion, the accuracy evaluation method of 3.3 evaluation index is changed to calculation precision, recall, F-score, and IoU. The corresponding tables 2, 3, 4 and experimental analysis are modified.

Point 12: For the results of the methods in Tables 2 and 3 against the proposed method, the authors should analyze whether the improvements are significant or not. I suggest performing a statistical analysis test and then according to the p-value, the authors can prove the significance of the improvements.

Response 12: I'm sorry for the trouble caused by the writing problem.

(1) The main purpose of the proposed method is to improve the automation of road extraction and reduce the number of manual input points on the premise of ensuring the accuracy. The selected comparison methods are all semi-automatic road extraction methods, and the accuracy of extraction results can be guaranteed by different human participation. Our advantage is to reduce the degree of human participation. Compared with the other three algorithms, the number of seed points is reduced by more than 40%, which highlights the higher degree of automation of our algorithm. According to your proposal, we added "on the premise of ensuring the accuracy, recall rate and other parameters to reach more than 98%, compared with other methods, the algorithm automation of this paper is improved by more than 40%." in 4. Discussion.

(2) According to the data, P-value is a random variable used to determine the hypothesis test results [1], and its value is related to the current sample observation value [2], and the calculation results will have certain randomness. The accuracy evaluation in this paper is based on the artificial ground truth, which is used to detect the accuracy of the whole image, and the evaluation results are more accurate. In addition, the accuracy evaluation of existing methods is almost based on the overall accuracy evaluation, such as reference [3][4][5][6].

[1] Murdoch, D.J.; Tsai, Y.L.; Adcock, J. P-Values are Random Variables. The American statistician. 2008, 62(3), 242-245.

[2] Wei, L.G.; Liu, G.J. Research on p-value teaching of hypothesis testing based on the idea of degree. Higher Education of Sciences. 2020, 153(05), 108-111.

[3] Lin, Y.N.; Xu, D.Y.; Wang, N.; et al. Road extraction from very-high-resolution remote sensing images via a nested se-deeplab model. Remote Sens. 2020, 12(18),293-293.

[4] Pan, H.; Jia, Y.; Lv, Z. An adaptive multifeature method for semiautomatic road extraction from high-resolution stereo mapping satellite images. IEEE Geosci. Remote Sens. Lett. 2019, 1-5.

[5] Lian, R.; Wang, W.; Li, J. Road extraction from high-resolution remote sensing images based on adaptive circular template and saliency map. Acta Geod. Cartogr. Sin. 2018, 47, 950–958.

[6] Dai, J.; Zhu, T.; Zhang, Y.; et al. Lane-level road extraction from high-resolution optical satellite images. Remote Sens. 2019,11(22), 2672.

Round 2

Reviewer 3 Report

There are some major comments that should be addressed by the authors. Otherwise, I can't accept the paper due to some reasons related to justification, proof, analyzing and additional experiments needed.

Point 2: A new section should be created entitled "Literature Review" or "Related Work"
It is possible to do it as if we see it is required. As an example, You can see the manuscript at the following link:
https://www.mdpi.com/2072-4292/13/7/1255/htm

Point 4: The point should be updated according to point 2.
Point 5: A new comment according to the answer to this point.
For contributions 2 and 3.
The authors should prove these two contributions mathematically or experimentally (means using metrics related to what you are going to prove).

Point 9: For this point, the authors should give the reference number nearest each method in Tables 3 and 4.

Point 12: This point is very important. The authors should do the experiments several times and record the results for each method used in the comparison, including your method; then, apply a statistical test to see whether the compared results of precision, recall, F1 are significantly improved or not.